# Complete telomere-to-telomere genomes uncover virulence evolution conferred by chromosome fusion in oomycete plant pathogens

Zhichao Zhang[1,2], Xiaoyi Zhang[1,2], Yuan Tian[1,2], Liyuan Wang[1,2], Jingting Cao[1,2], Hui Feng[3], Kainan Li[1,2], Yan Wang [1,2], Suomeng Dong [1,2], Wenwu Ye [1,2] ✉ & Yuanchao Wang [1,2] ✉

Variations in chromosome number are occasionally observed among oomycetes, a group that includes many plant pathogens, but the emergence of such variations and their effects on genome and virulence evolution remain ambiguous. We generated complete telomere-to-telomere genome assemblies for *Phytophthora sojae*, *Globisporangium ultimum*, *Pythium oligandrum*, and *G. spinosum*. Reconstructing the karyotype of the most recent common ancestor in Peronosporales revealed that frequent chromosome fusion and fission drove changes in chromosome number. Centromeres enriched with *Copia*-like transposons may contribute to chromosome fusion and fission events. Chromosome fusion facilitated the emergence of pathogenicity genes and their adaptive evolution. Effectors tended to duplicate in the sub-telomere regions of fused chromosomes, which exhibited evolutionary features distinct to the non-fused chromosomes. By integrating ancestral genomic dynamics and structural predictions, we have identified secreted Ankyrin repeat-containing proteins (ANKs) as a novel class of effectors in *P. sojae*. Phylogenetic analysis and experiments further revealed that ANK is a specifically expanded effector family in oomycetes. These results revealed chromosome dynamics in oomycete plant pathogens, and provided novel insights into karyotype and effector evolution.

Oomycetes include numerous devastating plant pathogens[1]. The well-known pathogens—namely, *Pythium*, downy mildews, and *Phytophthora*—presumably share a common origin, causing significant economic losses[2,3]. Effectors are key virulence factors or toxins that alter host physiology, contributing to the establishment of infection[4]. During the long-term arms race of host–pathogen interactions, effectors rapidly evolve to escape recognition of the plant immune system.

Effectors recognized by host resistance (R) proteins are known as avirulence (Avr) effectors[5]. In oomycetes, the most well-known Avr effectors are the RxLR (arginine–any amino acid–leucine–arginine) family; these genes are widely distributed, particularly in *Phytophthora*[6,7]. An understanding of the evolutionary trajectory of effectors is crucial for the development of long-term effective disease control strategies.

[1]Department of Plant Pathology, Nanjing Agricultural University, Nanjing 210095 Jiangsu, China. [2]Key Laboratory of Soybean Disease and Pest Control (Ministry of Agriculture and Rural Affairs), Nanjing Agricultural University, Nanjing 210095 Jiangsu, China. [3]Tobacco Research Institute, Chinese Academy of Agricultural Sciences, Qingdao 266101, China. ✉e-mail: yeww@njau.edu.cn; wangyc@njau.edu.cn

Effector evolution is tightly linked to genomic compartments. In filamentous pathogens, the most well-known model is the 'two-speed genome', involving gene-poor and repeat-rich compartments that are enriched in rapidly evolving effectors; and other genomic compartments are gene-rich and repeat-poor, with conserved genes[8–11]. The differences between these two compartments are presumably related to the expansion of *Gypsy* transposons[8]. This concept has been complemented by accessory chromosomes, sub-telomere regions and AT-rich isochores in fungi, which are also rapidly evolving regions[11–13]. Several high-quality fungal genome assemblies reveal that effector (including *Avr*) genes are found adjacent to telomeres or arranged in clusters at chromosome ends. Segmental duplications frequently occur between the core chromosome ends and accessory chromosomes. It has been proposed that effector variation in accessory chromosomes could be exchanged into core chromosomes, could contribute to pathogen evolution[14–16].

Pan-genome and comparative genomics analyses showed that structural variations (SVs; e.g., presence or absence variants, inversions, and translocations) significantly contribute to the evolution of functional genes[17,18]. Chromosomes also undergo larger-scale rearrangements, known as karyotype evolution. Polyploidy or aneuploidy karyotype variation in *P. infestans* contributes to resistance to metalaxyl[19]. Chromosome-level comparative genomics analysis revealed that two downy mildews, *Peronosclerospora sorghi* and *Peronospora effusa*, may have undergone four chromosome fusions (CFs)[20]. Generally, CF can contribute to reproductive isolation and speciation[21]. For example, two independent ancestral ape chromosomes fused into one chromosome in humans[22]. Because of multiple CF events, muntjac deer experienced a dramatic reduction in chromosome number during speciation[23]. CF is also involved in the early evolution of sex chromosomes[23,24]. Using CRISPR/Cas9-based technology in yeast and mice, CF was identified as an important pathway for the establishment of reproductive isolation[25–27]. Additionally, CF events may help novel species to adapt to divergent environments by altering the recombination landscape and combining previously unlinked loci[28]. We noted varying chromosomes in oomycetes, especially across genera, with unclear biological significance. Cytological observations suggest that *P. sojae* and *P. infestans* contain 12–14 and 8–12 chromosomes, respectively[29–32]. *P. plurivora* contains 18 chromosomes[33]. Among the downy mildews, *Bremia lactucae* and *Pe. effusa* may contain 19 and 17 chromosomes, respectively[34,35]. *Per. sorghi* is characterized by 13 chromosomes[20]. Clamped homogeneous electrical field gel electrophoresis resolved approximately 20 chromosomes for *Py. oligandrum* (41.5 Mb)[36]. Variation in chromosome number might hold particular significance in oomycete evolution.

Advances in long-read sequencing technologies have improved the ability to explore repeat-rich regions; genome assemblies have been considerably improved. In oomycetes, increasing numbers of high-quality assemblies have been obtained based on long-read sequencing, and some have achieved telomere-to-telomere (T2T)-level (*Pe. effusa*) or been close to chromosome level (*B. lactucae*, *Per. sorghi*, *P. plurivora* and *P. infestans*)[20,33–35]. T2T-level genome assemblies are still limited, and a comprehensive chromosome-level comparative genomic analysis in oomycetes is still insufficient.

In this study, we employed PacBio High Fidelity (HiFi) technology for highly accurate long-read sequencing and optimized assembling methods, successfully achieving T2T-level assemblies for four oomycete species: *P. sojae*, *G. ultimum*, *Py. oligandrum*, and *G. spinosum*. The ancestral genome reconstruction and comparative genomic analyses of Peronosporales revealed the phenomenon and molecular mechanisms of chromosome fusion and/or fission, which may contribute to the understanding of effector evolution. By integrating evolutionary hotspot regions with protein structure clustering in *P. sojae*, our study also reveals the contribution of secreted Ankyrin-repeat proteins in virulence, which may be a novel class of effectors.

## Results

### Complete T2T-level genome assembly of *P. sojae*

The obtained genome assembly *P. sojae* 2023 was 85.1 Mb and contained 12 contigs, including 11 T2T chromosomes and one single-telomere chromosome. Among the *P. sojae* genome assemblies *P. sojae* 3.0, 2019, and 2023, there was a good global collinearity and a consistent distribution of telomere repeat regions; however, the assemblies of *P. sojae* 2023 exhibited a prominently better completeness than *P. sojae* 3.0 (82 contigs, only one is T2T) and 2019 (70 contigs, none of them are T2T) (Fig. 1a and Supplementary Data 1).

The previous analysis of *P. sojae* 2019 revealed that centromere regions were enriched with *Copia*-like transposons and embedded in heterochromatin[37]. In *P. sojae* 2023, we identified a centromere in each of the 12 chromosomes (Supplementary Fig. 1). According to RNA-seq and assay for transposase-accessible chromatin (ATAC)-seq data, the transcription level and chromatin openness of these centromere regions were low, consistent with a state of heterochromatin (Fig. 1b). In a previous cytological study, *P. sojae* was estimated to have 12–14 chromosomes[38]. Therefore, our results suggested that the *P. sojae* 2023 assembly has achieved the T2T chromosome scale, with superior completeness, accuracy, and continuity.

### Effector evolution on chromosome landscape

A significantly greater abundance of repetitive elements was present around RxLR (effector) genes than around randomly selected genes (Supplementary Fig. 2a). According to comparisons of the type and coverage of 3-kb repetitive elements upstream and downstream of genes, most of the repetitive elements were identified as DNA transposons. *Sola3* was the most abundant repetitive element around RxLR genes, compared with randomly selected genes; *Helitron*, involved in rolling-circle duplication of Crinkler (CRN) effector genes in *P. infestans*[8], was also abundant around RxLR genes (Supplementary Fig. 2b). The identified tandem duplicated genes were significantly enriched in RxLRs, elicitins, as well as other pathogenicity-related families, such as secreted proteins, cell wall degrading enzymes, and transporters (Supplementary Fig. 2c). These results suggest that DNA transposons and gene tandem duplication may contribute to the evolution of effector genes.

RxLR genes tended to be clustered on chromosomes and associated with high SNP density. A total of 456 RxLR genes were identified, with 38 being novel RxLR genes (sequence identity <80%) in *P. sojae* 2023 as compared to previous study[39]. Among these novel identified genes, 17 were located in near-telomere regions; their corresponding genomic fragments were not assembled or subjected to SVs in *P. sojae* 2019 (Supplementary Fig. 3 and Supplementary Data 2). Compared with the *Avr* genes identified in *P. sojae* strain P6497, *Avr1b* and *Avr1d*—located in an 11-kb highly divergent region and a 3-kb highly divergent region, respectively—were absent in *P. sojae* 2023 (Supplementary Fig. 4); *Avr1a*, *Avr1c*, *Avr3a/5*, *Avr3b*, and *Avr3c* mainly contained non-synonymous mutations (Supplementary Fig. 5). The mutation of these *Avr* genes may be responsible for variations in virulence, as previously reported[40]. These results suggest that both SNPs and SVs play critical roles in promoting the evolution of effectors.

### Karyotype evolution driven by ancestral chromosome fusion and fission

To explore the evolution of oomycete chromosomes, we reconstructed a phylogenetic tree which comprised 38 representative oomycete species, including 12 downy mildews species, 18 *Phytophthora* species, 1 *Phytopythium* species, 4 *Globisporangium* species, 2 *Pythium* species, and *Saprolegnia parasitica* (as an outgroup). Based on the T2T assemblies of *P. sojae* 2023, three newly generated Pythiaceae species (*G. ultimum*, *Py. oligandrum*, and *G. spinosum*) in this study, and five previously reported species, we observed that 18 chromosomes are in the three Pythiaceae species, while significant

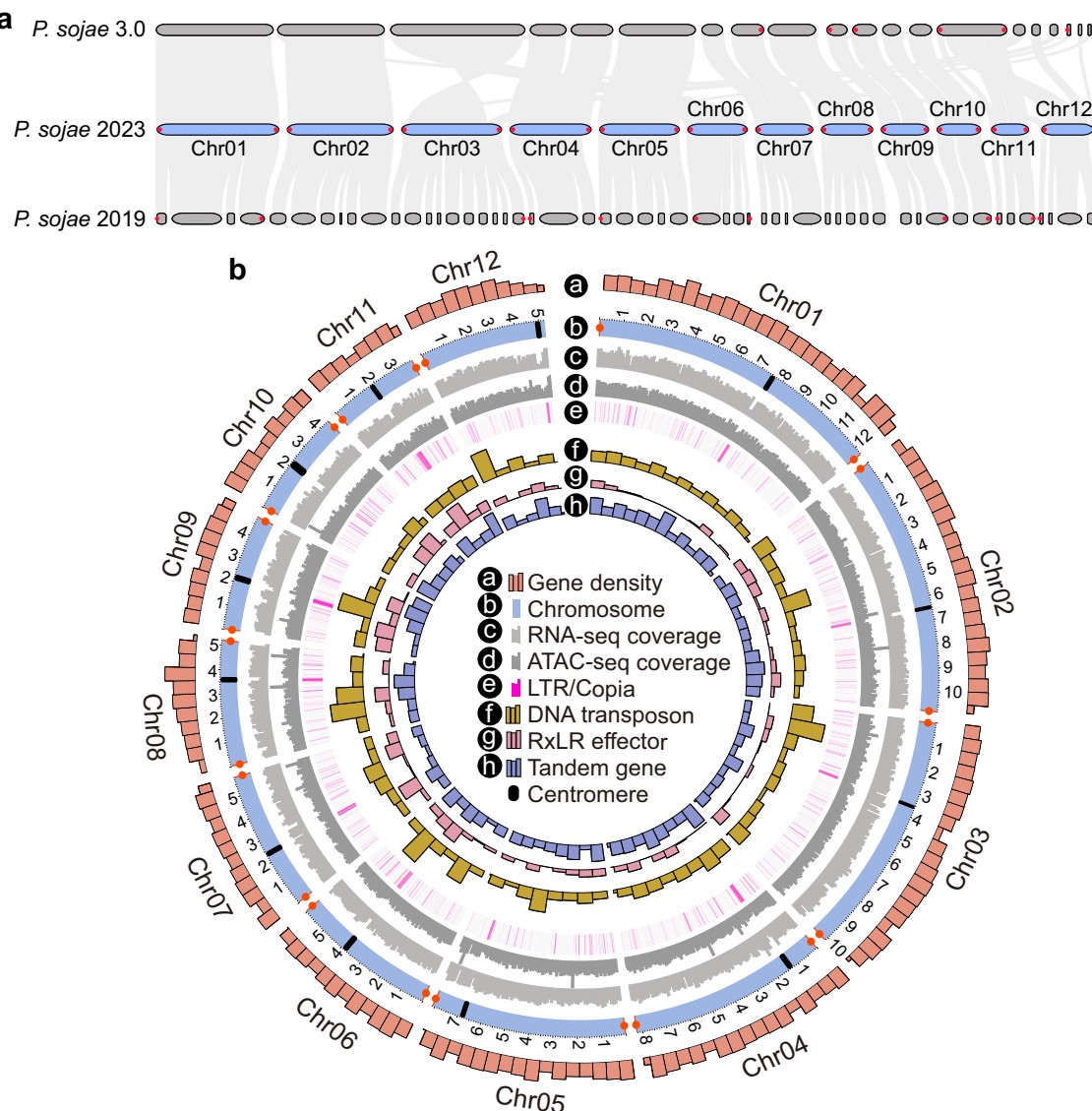

**Fig. 1 | Characteristics of the *P. sojae* 2023 T2T genome assembly. a** Comparison of collinearity and telomere distribution among *P. sojae* 2023, *P. sojae* 3.0, and *P. sojae* 2019. The *P. sojae* 2023 genome was assembled in this study. **b** Visualization of *P. sojae* 2023 T2T assembly characteristics. Circular layers represent the following (from outside to inside): **a** gene density, total number of genes in 500-kb windows; **b** chromosome, minimum scale 100 kb. **c, d** RNA-seq and ATAC-seq

coverage were visualized according to number of reads (normalized using base-2 logarithms) in 100-kb windows. **e** Pink lines represent distribution of *Copia*-like transposons. **f** Brown bar represents number of DNA transposons in 1-Mb windows. **g** Pink bar represents number of RxLR effector genes in 1-Mb windows, and h. blue bar represents number of tandem duplication genes in 1-Mb windows. Source data are provided as a Source Data file.

variations of chromosome number (from 12 to 19) are among different lineages in the Peronosporales (Fig. 2a).

Variations in the number of chromosomes indicate that chromosome fusions, fissions and/or losses may occur in this lineage. To investigate the evolutionary trajectory behind varying chromosome numbers, we further compared the chromosome-level assemblies of nine oomycete species, spanning most of the evolutionary branches. The synteny of the three Pythiaceae species showed the stability of the karyotype, and only a few (1 or 2) rearrangements occurred (Supplementary Fig. 6). Five chromosomes (Chr01-05) may have undergone fusions in *P. sojae*, while the other chromosomes (Chr06-12) still retained the ancient karyotype (Fig. 2b). Additionally, *Pe. effusa* had three chromosomes (Chr09, 11, and 12) aligned to *P. sojae* Chr02, while *P. plurivora* had five chromosomes (Chr09, 04, 14, 17, and 06) aligned to Chr02, indicating the presence of several new fusion and/or fission events (Fig. 2b). The fact that *P. plurivora* had higher chromosome numbers than *P. sojae* (18 *vs.* 12) may be attributed to more fissions.

Reconstruction of ancestral karyotypes further suggested that fusions were the main variants of *P. sojae*, and *P. plurivora* may have eight fusions and seven fissions since diverging from the ancestor node A0, forming 18 chromosomes (Fig. 2c and Supplementary Fig. 7). Together, our results suggested that the karyotype of oomycetes was dynamic, mainly shaped by multiple chromosome fusion and fission events, leading to significant differences in chromosome number among species.

## Centromere dynamics contribute to karyotype evolution

To investigate the potential molecular mechanisms of chromosome fusion and/or fission, we conducted chromosome-level collinear analyses between *P. sojae* and *P. plurivora*, finding that *P. plurivora* underwent more fusions and fissions. Macro/micro-synteny results between *P. plurivora* and *P. sojae* indicated that almost all fusions or fissions originated from or were attributed to the centromere regions, reflecting the general significance of centromere-mediated

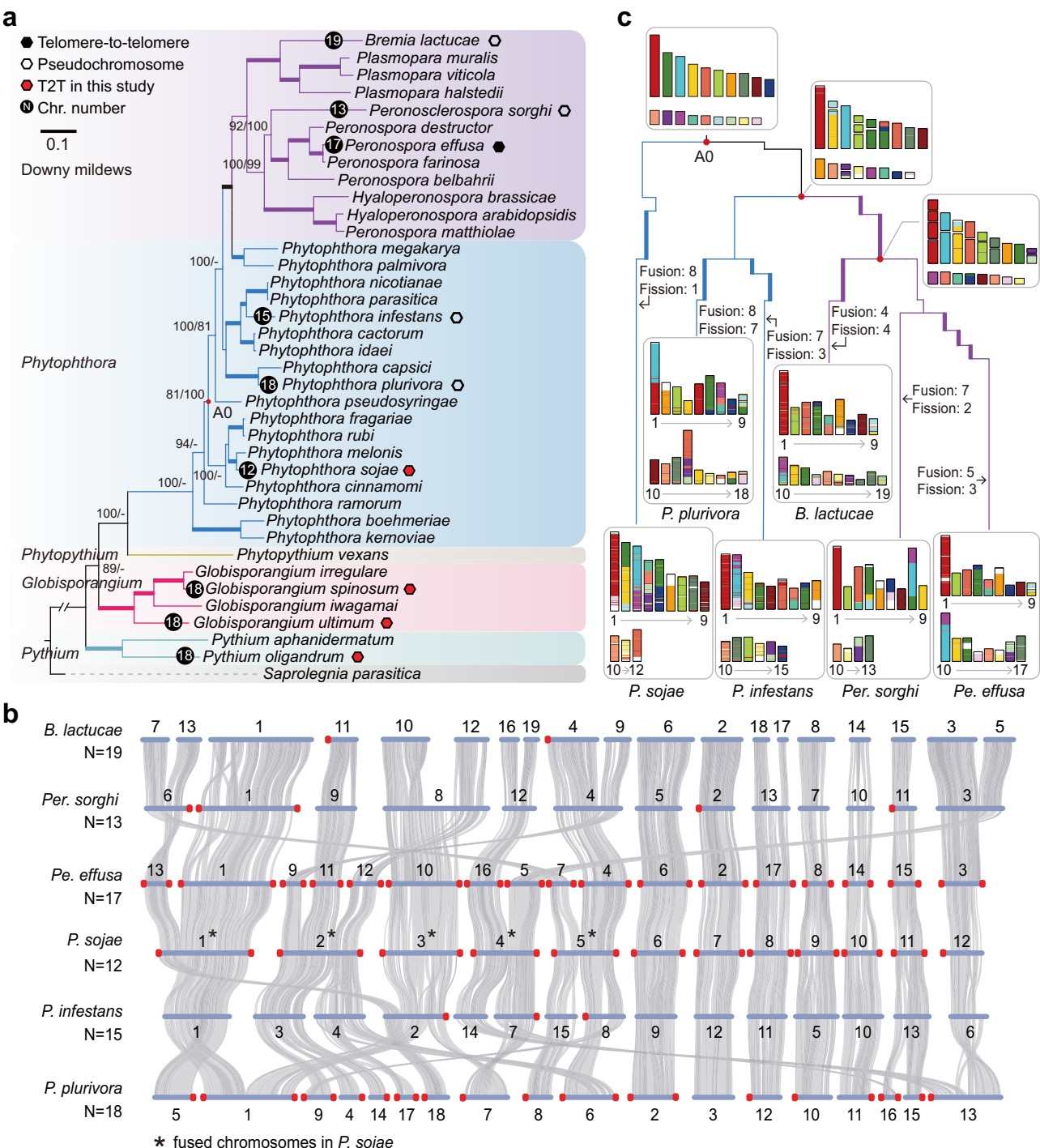

**Fig. 2 | Reconstruction of Peronosporales ancestral chromosomes. a** A total of 38 species were used to construct the phylogenetic tree, using *S. parasitica* as an outgroup. Bootstrap value for maximum likelihood and minimum evolution inference is added on the branches in their respective order. A minus sign indicates a lack of support for consistent topology. Thickened branches represent '100/100' in both analyses. A0 represents one most recent common ancestor node of *Phytophthora* and downy mildews species, and the red dot indicates the ancestral node. **b** Chromosome-level synteny analysis was performed between representative three downy mildews (up), and three *Phytophthora* species (down). **c** The 27 species belonging to node A0 were used for the reconstruction of ancestral chromosomes. In order to show the dynamics of fusion and fission, chromosome-level modern species were colored based on the karyotypes of the ancestral node A0. Events such as chromosome fusions and fissions that occurred during the evolution from A0 to modern species are indicated in the figure. Source data are provided as a Source Data file.

chromosome fusions or fissions (Fig. 3 and Supplementary Fig. 8). Moreover, we compared *P. sojae*, *G. ultimum*, and *Pe. effusa*, all of which have relatively complete levels of T2T assembly. Numerous inverted variations were observed between *G. ultimum* and *P. sojae*. Fused sites were distributed within these inverted regions. It was observed that those regions surrounding centromeres PsCEN1, GuCEN1, GuCEN6, GuCEN11, and GuCEN16 maintained conserved collinearity at the fusion sites (Supplementary Fig. 9). Centromere positions among *G. ultimum*, *Pe. effusa*, and *P. sojae* were relatively conserved. Nonetheless, multiple chromosomes were fused into a

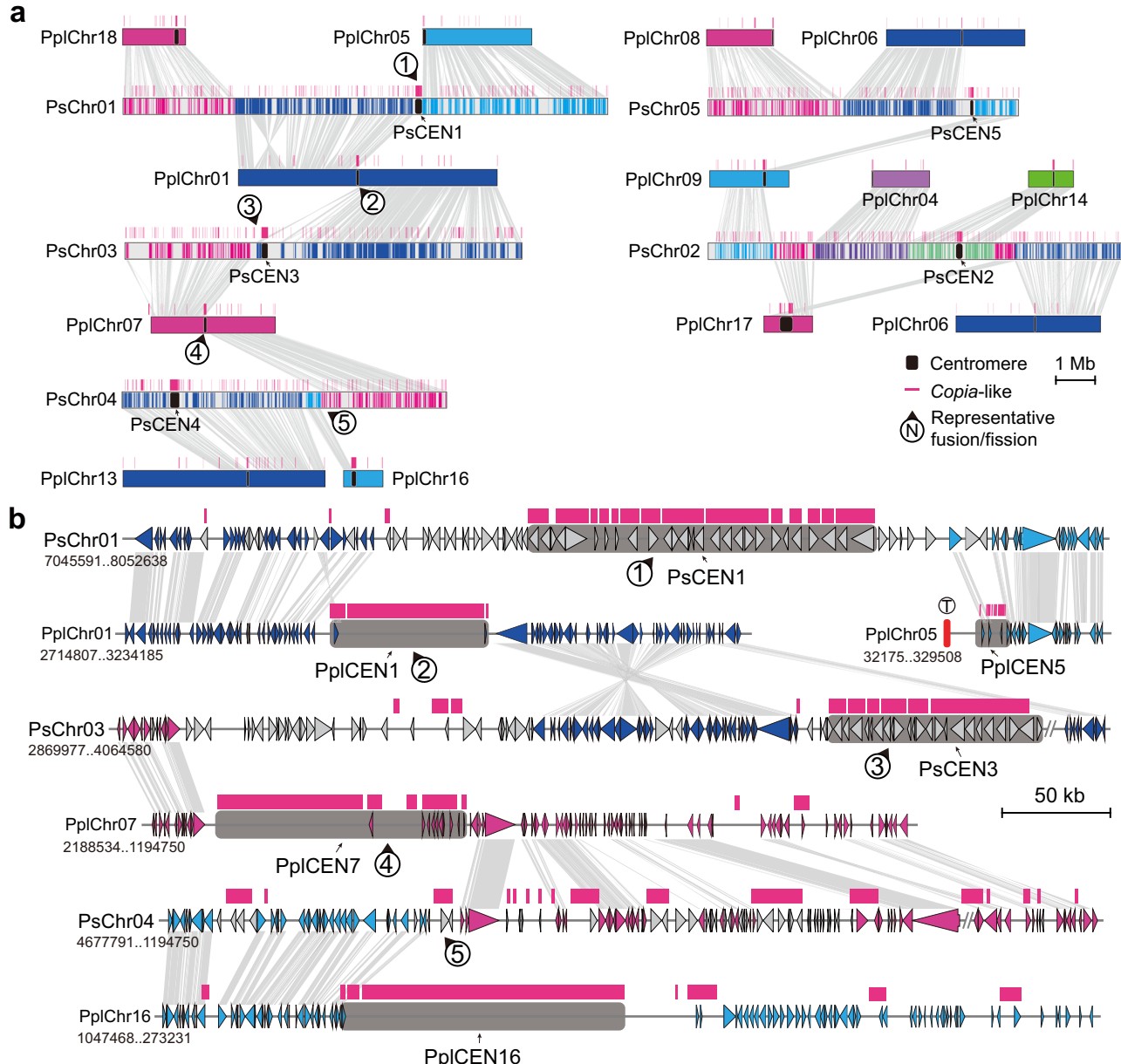

**Fig. 3 | Centromeres drive chromosome fusion or fission events. a** The figure shows the chromosomes fusion or fission occurred between *P. sojae* and *P. plurivora*. The arrow points to the chromosome fusion or fission region. The gray lines connecting chromosomes represent regions of conserved synteny. CEN represents centromere and red lines above chromosomes represent the distribution of *Copia-like* elements. **b** Micro-synteny in the regions of fusion or fission on *P. sojae* Chr01, Chr03, and Chr04, corresponding to the left side of Fig. 3a. Chr02 and Chr05 are shown in the Supplementary Fig. 8. Source data are provided as a Source Data file.

single chromosome in *P. sojae*, and the excessive centromeres on the fused chromosomes may be lost. This was manifested by the absence of *Copia*-like sequences in these regions (Supplementary Fig. 10). Since dicentric or multicentric activated chromosomes are unstable, extra centromeres are either epigenetically silenced or deleted to reduce their instability, thereby producing functionally monocentric chromosomes that are properly separate during cell division[41]. These results suggested that centromeres enriched with *Copia*-like elements in oomycetes played an important role in promoting chromosome fusion and fission.

## Fused and non-fused chromosomes exhibit distinct evolutionary features

In *P. sojae*, fused chromosomes (FCs) occupied a large portion (5/12). Compared with non-fused chromosomes (nFCs), chromosome size and gene density were significantly higher, while SNP or effector gene density, and repeat content were lower in FCs (Fig. 4a). Repeat content was inversely proportional ($r = -0.66$) to chromosome size and proportional ($r = 0.63$) to intergenic distance (Fig. 4b). Neither the intact *Copia*-like nor *Gypsy* transposons show significant differences between fused and non-fused chromosomes (Supplementary Fig. 11), but the insertion time distribution of *Copia*-like on fused chromosomes were more recent than non-fused chromosomes (Fig. 4c). Given the essential role of *Copia*-like transposons in centromeres, we speculated that the dynamic changes of insertion time may be related to chromosome fusion.

Interestingly, the distribution of RxLR genes in FCs was closer to telomere than in nFCs (Fig. 4a). Cross-chromosome segmental duplications often occurred at near-telomere regions, and abundant pathogenicity-related genes were enriched in these regions, including

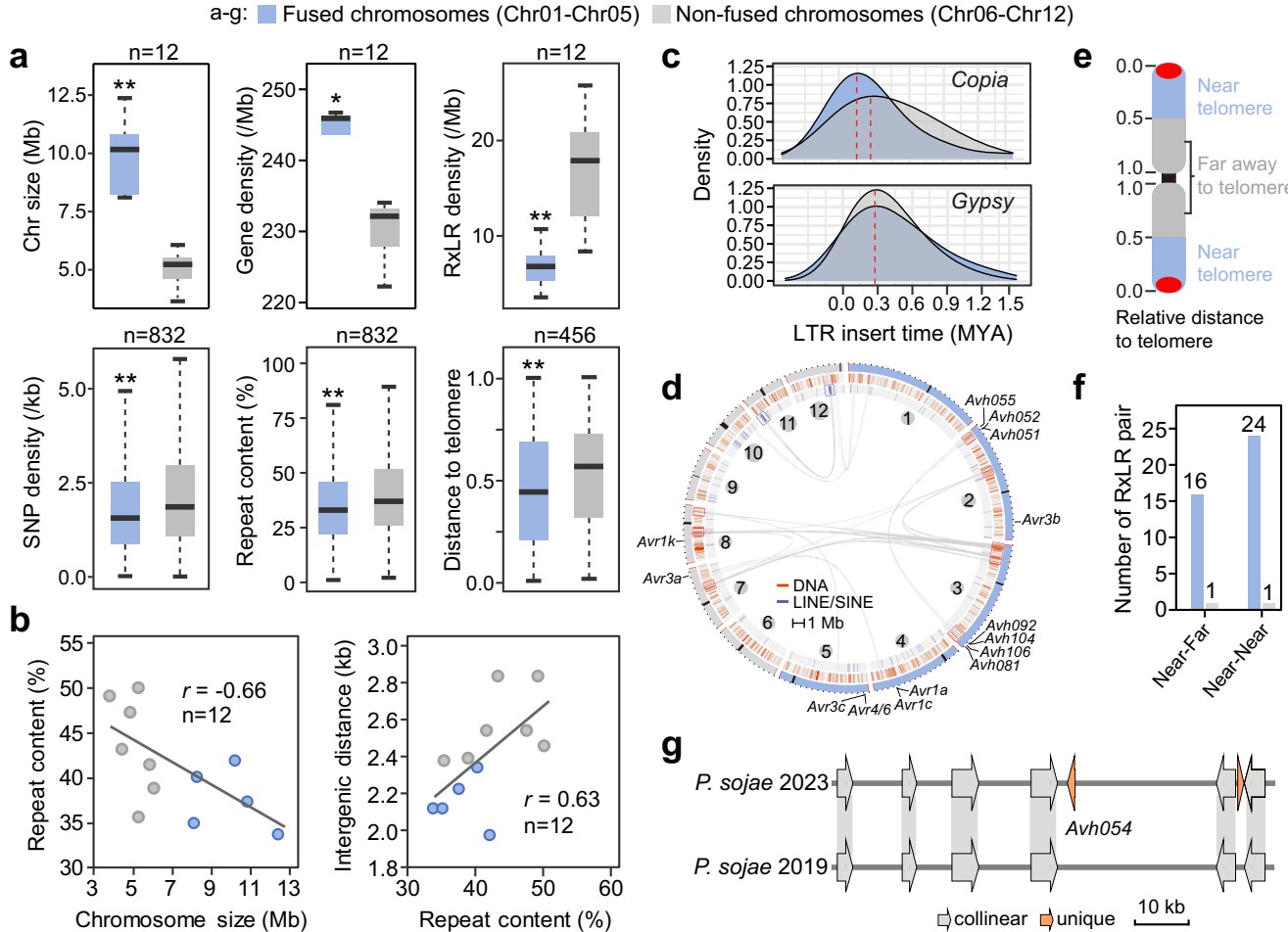

**Fig. 4 | Comparison of fused and non-fused chromosome characteristics.**
**a** Comparisons of chromosome size ($p = 0.0052$), gene density ($p = 0.05$), repeat content ($p = 0.00058$), SNP density ($p = 0.01$), RxLR gene density ($p = 0.01$), and the distance of RxLR genes relative to telomere ($p = 2.5e−09$) between fused (blue) and non-fused (gray) chromosomes. *P*-values were calculated by two-sided Wilcoxon tests. The center line in box plots indicates the median, the box outlines the 25th and 75th percentiles, and the whiskers extend to 1.5 times the interquartile range beyond the box edges. **b** Correlation of chromosome size, repeat content, and intergenic distance; *r*, Pearson correlation coefficient. **c** Density plot represents the distributions of intact *Copia* and *Gypsy* elements in specific insert times.

**d** Comparison of collinearity among chromosomes of *P. sojae* 2023. *Avr* genes are indicated in chromosomes. Minimum scale is 500 kb. **e** Model of relative distance from gene to telomere. **f** Number of inter-chromosome duplicated RxLR genes located in near-telomere regions. Near-Far, one of the RxLR pair is located in the near-telomere region; Near-Near, both of the RxLR pair are located in the near-telomere region. **g** Micro-synteny between *P. sojae* 2023 and *P. sojae* 2019. The coordinates are Chr02:348308-404549 and c12:1349752-1406164, respectively. All gene models are corrective in corresponding genomes. Source data are provided as a Source Data file.

RxLR effectors (Supplementary Fig. 12). Meanwhile, these regions occupied DNA transposons or LINE/SINE retrotransposons, suggesting that these transposons may mediate segmental duplications (Fig. 4d and Supplementary Fig. 13). Notably, five of the seven identified *P. sojae Avr* genes (*Avr1a*, *Avr1c*, *Avr3c*, *Avr4/6*, and *Avr3a/5*) were found in near-telomere regions (Fig. 4d, e). Nearly all RxLR genes that were duplicated in near-telomere regions were found in FCs (Fig. 4f). *Avh054*, a RxLR gene with insertion/deletion polymorphism resulting in a truncated protein, were found at near-telomere region of the FC Chr02 (Fig. 4g, Supplementary Figs. 14 and 15). Segmental duplication in the proximal region of telomeres facilitates gene flow between chromosomes and may contribute to effector evolution. Together, these findings indicated that evolutionary features differed between FCs and nFCs, which presumably function as two compartments within the genome.

## Chromosome fusion facilitates the adaptive evolution of effectors

Chromosome fusion may reshape chromosomal architecture, leading to gene loss and novel gene combinations that confer advantages in the virulence evolution of *P. sojae*. To explore this possibility, we defined evolutionary breakpoint regions (EBRs) based on past reports[42] (Supplementary Fig. 16a). SNPs and CNVs in EBRs, sub-telomere regions, and genome-wide averages showed no significant differences, which may be explained by contributing to the stability of the genome after speciation (Supplementary Fig. 16b–d). Genes enriched pathways in EBRs were related to DNA metabolism, integration, and nuclease activity (Fig. 5a), which may be involved in biological processes like chromosome recombination, repair, and stability. In addition, pathways related to pectate lyase (PL) and extracellular regions were also significantly enriched (Fig. 5a). In the EBRs of Chr04, we identified nine tandemly duplicated genes of the PL family based on functional domain prediction and protein structural comparison (Fig. 5b and Supplementary Fig. 17a). Interestingly, the PL family contained 62 members, showing a significant expansion in *P. sojae* (Supplementary Fig. 17b). The nine tandemly duplicated PL genes in the EBRs were specifically expanded in *P. sojae*, and exhibited relatively high expression levels during infection (Fig. 5c and Supplementary Fig. 17c).

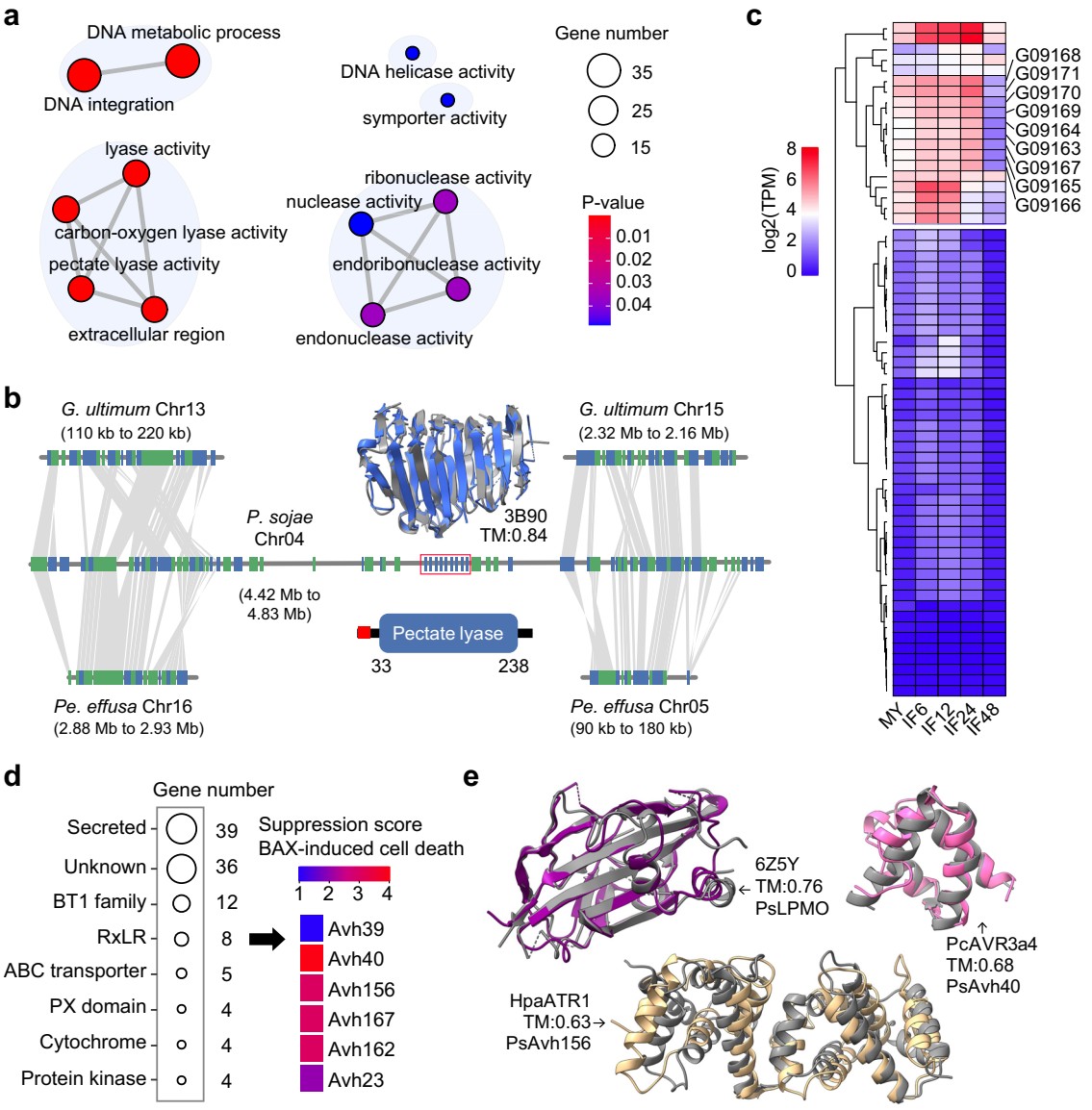

**Fig. 5 | New genes and effector evolution in fused chromosomes. a** GO enrichment analysis is conducted in fused regions. The lines represent functionally related GO terms. *P*-values were calculated by the one-sided hypergeometric distribution test, with FDR adjustments for multiple comparisons. **b** PDB ID (3B90) represents the reference template structure of a pectate lyase, shown in white. The AlphaFold2 predicted model is shown in blue. TM indicates the TM-score calculated from the TM-align structural alignment. The predicted domains by SMART are shown at the bottom, with red color representing the signal peptide region, and the numbers indicating the start and end positions of the domain. **c** The expression patterns of 62 PL family genes identified in *P. sojae* 2023 are presented as a heat map. Nine tandem genes are marked with asterisks, and their gene IDs are presented on the right side. The TPM value is normalized using base-2 logarithms. **d** The left panel shows the identified secreted proteins and RxLR proteins in fused regions, and the middle panel presents the BAX-induced cell death suppression score, as reported in Wang et al. **e** The gray structures represent LPMOs and RxLR PDB templates, while other colors correspond to the query protein structures predicted by AlphaFold2. The orange font indicates TM-score, calculated by TM-align. Source data are provided as a Source Data file.

In addition, we also identified 39 secreted proteins in other EBRs, including a member of the copper-dependent lytic polysaccharide monooxygenases (LPMOs) family identified by structural alignments. LPMOs can cleave pectin of the plant cell wall, contributing to the infection, as reported in *P. infestans*[43]. Eight RxLRs emerged in EBRs, six of them have been reported to inhibit BAX-induced cell death[44], and PsAvh23 can aid in *P. sojae* infection by inhibiting H3K9 acetylation in soybean[45]. PsRxL467 is a predicted RxLR effector with an additional 3'-5' exonuclease domain at C-terminus (Fig. 5d). PsAvh40 and PsAvh156 displayed highly structural similarities with the functional known RxLR effectors *Hyaloperonospora arabidopsidis* ATR1 and *P. capsici* AVR3a4, respectively (Fig. 5e and Supplementary Data 3). These results further suggested that CFs may be an important mechanism for the evolution of pathogenicity and demonstrate the diversity of virulence evolution in the oomycete plant pathogens.

## Ancestral genome reconstruction assists in identifying novel effectors

In *P. sojae* 2023, approximately 70% (12542/18311) genes were found to derive from conserved and contiguous ancestral regions, while others may have either emerged or rearranged during chromosome evolution. Most RxLR (372/456) and CRN (138/140) effector genes are located in these flexible regions, indicating that these regions have been hotspots of evolution since diverging from the ancestral chromosome (Supplementary Data 4). Effectors evolve rapidly, which often results in low sequence conservation; however, they may still maintain relatively conserved structures[46,47].

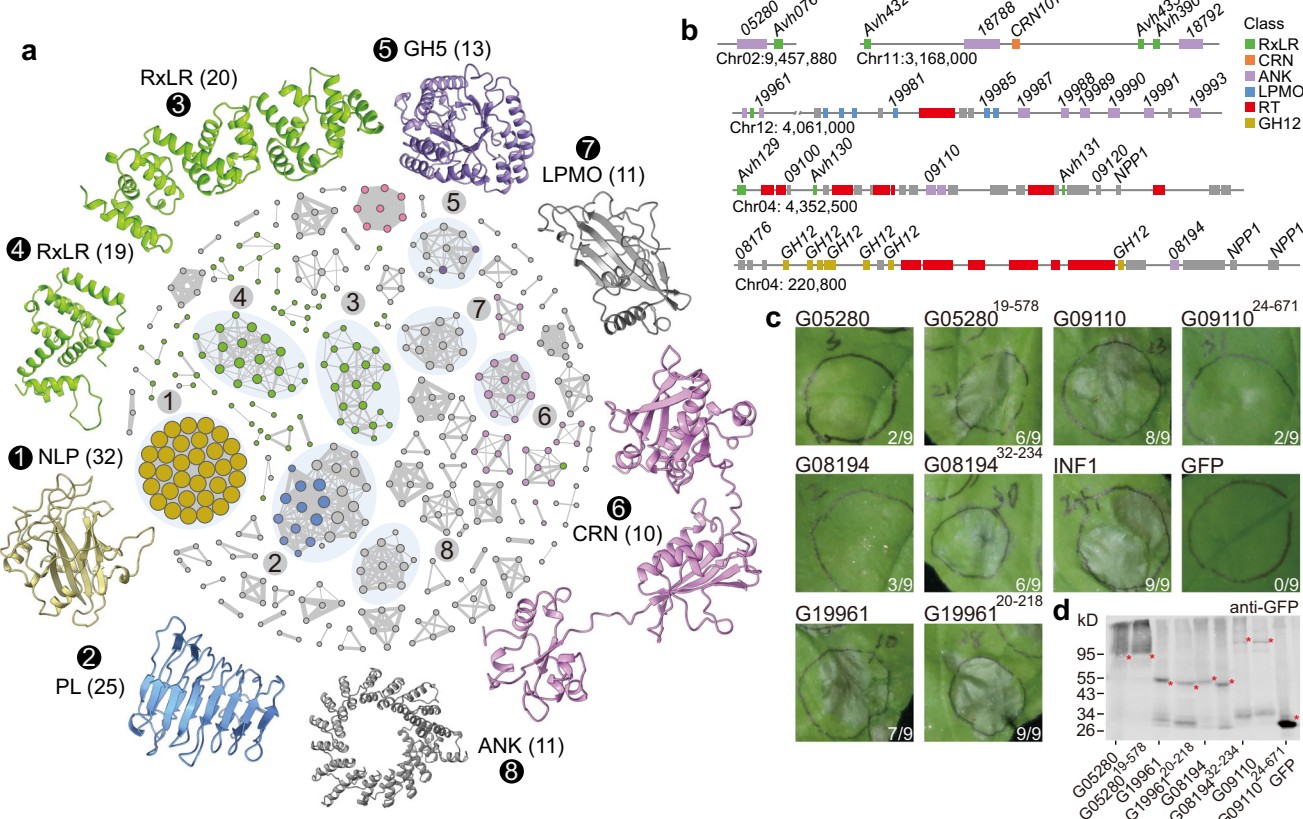

**Fig. 6 | Prediction of novel effector families. a** Protein structure prediction and clustering in hotspot regions enrich known and putative effector gene families. Gray represents uncharacterized and other gene families. The number in the circle represents the cluster ID. The number in parentheses represents the number of proteins. Two additional clusters were predicted as candidate effector families; both are marked as a gray structure. **b** The location of ANK repeats, LPMOs, RxLR, CRN, GH12 (Glycosyl hydrolase family 12), and RT (Retrotransposon) is shown on chromosomes. The gray box represents other genes. **c** Cell death induced by transiently expressing ANK effector candidates in *N. benthamiana* leaves. The experiments were performed with signal peptide removal and retention, respectively. Leaf infiltrations were imaged at 4 days after infiltration. INF1 served as a positive control. GFP was used as a negative control. The experiments at least replicated 9 times, yielding similar results. **d** Immunoblot analysis of protein levels in *N. benthamiana* leaves after transient expression of ANK effector candidates. All experiments were repeated three times with similar results. Source data are provided as a Source Data file.

Based on the above understandings, we performed protein structure-based clustering of genes in evolutionarily hotspots regions to identify potential novel effector families. We found that the well-known pathogenicity-related gene families, e.g., RxLR, CRN, NLP, and PL, formed large clusters with conserved structural features (Fig. 6a). Interestingly, RxLR families display more divergent sub-clusters, yet possess core structural units, which may have arisen from structural rearrangements, consistent with earlier findings[47]. Effectors usually have repeat-containing domains that easily change in protein size and accumulate mutations without affecting their original function[48]. In addition, we identified two uncharacterized large clusters. Cluster 7 had an unknown function based on sequence annotation but shared a homologous structure with the LPMOs family, demonstrating the feasibility of our identification method. Cluster 8 represented the Ankyrin repeat-containing protein (ANK) family is composed mostly of alpha-helices and internal domain repeats, similar to the architecture of WY/LWY domain in RxLR effectors (Fig. 6a). ANK family is relatively abundant in *S. parasitica* and *G. ultimum*[49,50]. The ANK genes were also up-regulated in the germinated cysts of *S. parasitica*, indicating its importance in the initial stage of host colonization[51]. However, the role of the secreted ANKs in plant filamentous pathogens remains poorly understood.

Compared to fungi and bacteria, a significant expansion of ANK members was observed in oomycetes. Particularly, the number of ANK members per species, the number of ANK repeat per protein, and the proportion of ANK proteins with signal peptide were all notably higher in necrotrophic and hemi-biotrophic oomycete pathogens (Supplementary Fig. 18a), implying their potential involvement in pathogenesis. ANK repeat units, consisting of dual alpha-helices and a central loop, exhibited conserved amino acid residues among oomycetes, diatoms, fungi, plants, and bacteria, suggesting a shared ancient origin (Supplementary Fig. 18b). Hydrophilic residues were distributed on the protein surface of ANK repeat unit, while hydrophobic residues were positioned internally, with three leucine (L) residues forming hydrophobic interactions that may stabilize the two alpha-helical structure (Supplementary Fig. 18c). ANK members were classified into five groups, in which the secreted ANKs of oomycetes were uniquely expanded in G5 (Supplementary Fig. 18d), further suggesting their divergent evolution in oomycete plant pathogens. Meanwhile, these secreted ANKs exhibited more significant conservation in structure than in sequence (Supplementary Fig. 19).

## ANKs may be a class of novel effectors in *P. sojae*

The tandem arrangement of ANK, RxLR, CRN, LPMO, and GH12 family effectors in genomic regions indicated that there may be an evolutionary and/or functional correlation (Fig. 6b). To further reveal the role of secreted ANK proteins in plant-microbe interactions, we conducted preliminary experiments on twelve members (Supplementary Data 5). Four of them were observed to induce cell death when expressed in *N. benthamiana* leaves (Fig. 6c). Immunoblot analysis

suggested that the protein sizes were as expected (Fig. 6d). To more directly determine the contribution of ANK genes in *P. sojae* virulence, two independent knockout lines (Line1 and Line2) of *G05280* were generated (Supplementary Fig. 20a–c). These *P. sojae* mutants both displayed weakened pathogenicity to soybean (Supplementary Fig. 20d, e). Thus, this ANK gene is required for *P. sojae* virulence. Notably, G09110 required a signal peptide to induce cell death, possibly functioning in plant apoplast (Fig. 6c). Plant cell surface receptors typically interact with Leucine-Rich Repeat Receptor-Like Kinase (LRR-RLK) BRI1-ASSOCIATED KINASE-1 (BAK1) or LRR-RLK SUPPRESSOR OF BIR1-1 (SOBIR1) to initiate immune responses. To investigate the potential role of ANK family in plant immunity, we expressed the candidate apoplastic effector G09110 in the wild-type, as well as in *BAK1*- or *SOBIR1*-knockout lines of *N. benthamiana* (Supplementary Fig. 20f). The induced cell death was compromised in the knockout mutants, suggesting a potential recognition of G09110 by plant cell-surface receptor-like proteins. Taken together, our results provide preliminary evidence that secreted ANKs may be a class of novel effectors in *P. sojae*, which strengthens our understanding to plant-*Phytophthora* interactions.

## Discussion

Using PacBio HiFi sequencing technology, we obtained T2T-level genome assemblies for *P. sojae*, *G. ultimum*, *Py. oligandrum*, and *G. spinosum*. Previously, *Pe. effusa* had reached the T2T level in oomycetes[35]. These assemblies will serve as important resources for comparative genomics, functional genomics, and population genetics, as well as references for other oomycete genome assemblies.

Karyotype evolution associated with chromosome fusion reflects genomic plasticity. Our results find that FCs and nFCs show different evolutionary features, especially in repeat content and effector, suggesting CF may contribute to the virulence evolution. Our T2T assembly also confirms RxLR clusters found in *P. infestans*[19] and reveals the near-telomere distribution of RxLR in *P. sojae*. Interestingly, we found that near-telomere duplication of RxLR in *P. sojae* is often located on FCs and may be mediated by DNA transposons. A similar mechanism occurs between the core chromosome and the accessory chromosome in wheat blast fungus[14]. Near-telomere segmental duplication may assist rapid evolution of RxLR effectors. Of the nine identified *Avr* genes, six were located in the near-telomere region, including the absent variant *Avr1d*.

Karyotype evolution driven by chromosome fusion is crucial for speciation and adaptation[25–27], yet its effects on filamentous plant pathogens remain unexplored. Here, we present solid evidence of chromosome number evolution in oomycetes and propose its contribution to the virulence evolution of *P. sojae*. Karyotype reconstructions based on different tree topologies show slight variations, linked not only to differences in the topology of phylogenetic trees but also to the impact of limited chromosome-level assemblies. Further chromosome-level assemblies and exploration of the relationships of downy mildews within Peronosporales remains necessary. Reconstruction using the topology tree shown in Fig. 2a better fits the number of chromosome fusions/fissions observed in chromosome synteny of oomycetes, leading to a more parsimonious reconstruction and better support. However, the fragmented nature of assemblies for most species may disrupt continuous ancestral regions without enough supporting evidence. Similar to the conclusion of previous study[35], the reconstructed ancestral karyotype proposed that there may be 18-19 ancestral chromosomes, which is close to the number of 18 conserved chromosomes in Pythiaceae. In the future, more chromosome-level assemblies will help improve the accuracy of reconstruction. Using AlphaFold2, we can identify conserved patterns of rapidly evolving effectors structurally and theoretically predict novel effector families by these hotspots genes based on structure clustering. Combining ancestral genome reconstruction with AlphaFold2 refined structure predictions and reduced computational demands. The identified ANK repeats family expands over 300 members in *P. sojae*, characterized by a varying number of repetitive unit of two α-helices, similar to the WY/LWY repeats of RxLR effectors, but the function of secreted ANKs are still unknown in oomycetes. Our study reinforced previous findings[49–51], highlighting the significance of the ANK family in oomycetes. In bacteria, ANK family members act as intracellular effectors, suppressing plant immunity by interacting with DNA or proteins[52]. Four ANK proteins could trigger cell death in *N. benthamiana* in this study. These results suggested that ANK proteins likely represent a class of conserved effectors in plant-microbe interactions. The ANK gene *G05280* was required for virulence. Further functional studies of ANK are interesting, but beyond our scope. We plan to further promote ANK-related research in the future and knock out more ANK genes to verify the contribution of ANK to pathogenicity and explore its interaction mechanism. G09110 induced co-receptor SOBIR1-dependent cell death and was potentially recognized by cell surface receptor-like proteins. Subsequent research is essential to identify the specific immune receptors and explore the immune phenotype in depth. In the future, we plan to further elucidate the functions and mechanisms of secreted ANK family proteins in *P. sojae*.

In conclusion, we generated multiple T2T genomes assemblies for oomycetes and demonstrated the widespread chromosome fusion and fission phenomenon in oomycetes, which further expanded the findings of Fletcher K et al.[20]. We unveiled the role of *Copia*-like transposons in karyotype evolution and proposed how CF events promote virulence evolution in *P. sojae*, based on a pioneering analysis of multi-omics data from *Pythium*, downy mildews, and *Phytophthora*. Our study also confirmed that ANK is a specifically expanded effector family in oomycetes. These findings offer a novel insight into the field of plant-microbe evolution. In the future, more T2T assemblies of oomycetes and comparative genomics analyses would be useful to further confirm our findings.

## Methods
### Sample preparation, DNA extraction, and PacBio HiFi sequencing

The sequenced *P. sojae* JS2 (termed 'P. sojae 2023' for its genome assembly), *G. spinosum* Pys2-2, and *Py. oligandrum* Po34 strains were isolated from soybean roots sampled in Nanjing, Jiangsu Province, China; and the *G. ultimum* Pyu18-6 strain was isolated from soybean roots sampled in Zibo, Shandong Province, China. *P. sojae* was inoculated on 10% V8 plates and cultured in the dark at 25 °C. After 5 days of culture, 10 mycelial blocks of 4 × 4 mm were cultured in V8 liquid medium at 25 °C in the dark for 5 days. Then, the mycelium was filtered through sterile gauze, the excess moisture of the mycelium was dried with absorbent paper, and the sample was stored in liquid nitrogen (~2 g per sample). *G. spinosum*, *Py. oligandrum*, and *G. ultimum* strains recovered activity after 2 days of culture, and the mycelium was collected after 3 days of growth, and other culture conditions were identical to those for *P. sojae*.

For DNA extraction, 0.3–3 g of mycelium was ground into powder with liquid nitrogen and transferred to a 50-mL centrifuge tube; 5–25 mL of 2% Lysis Buffer 1 were added to the tube and mixed by shaking. The mixture was incubated at 65 °C for 120 min, then centrifuged at room temperature for 10 min. The supernatant was transferred to a clean centrifuge tube and 2/3× to 1× of the supernatant volume of isopropanol was added and slightly mixed by inversion; the mixture was then incubated at −20 °C for ≥2 h. After incubation, the tube was centrifuged at room temperature for 10 min to form a clear pellet at the bottom of the tube. The supernatant was removed; the pellet was washed twice with 750 μL of 75% ethanol by gently shaking the tube or pipetting the pellet up and down, then transferring into a 1.5-mL centrifuge tube. Subsequently, the tube was centrifuged at room temperature for 10 min and the supernatant was discarded; the pellet

was centrifuged at $12,000 \times g$ for 120 s at room temperature, then air-dried. After the pellet had dried, EB solution was added to the tube according to the size of the DNA precipitate, then incubated at 37 °C until the pellet had completely dissolved. The DNA was then fragmented using the Megaruptor system to a mean size of 15–20 kb. The adapter ligation reaction system was prepared such that the dumbbell-shaped adapter was connected with DNA to form a SMRTbell with hairpin structure. Qualified libraries were sequenced on the PacBio Sequel II platform. The library construction and PacBio HiFi reads were generated by the Beijing Genomics Institute (https://www.bgi.com/).

### De novo T2T genome assembling and assembly assessments
Cutadapt v2.6 (https://github.com/marcelm/cutadapt) was used to identify and trim adapter sequences, primers, poly-A tails, and other types of redundant sequences from raw sequenced reads. PacBio HiFi reads were assembled with Hifiasm v0.14[53] and HiCanu v2.0[54] using default parameters. We filtered out contigs with low GC content (<25%) and mitochondrial genome DNA from our analysis. By aligning contigs generated by Hifiasm with those from HiCanu, we assembled complete chromosomes and appended telomere sequences[35].

To assess the completeness and contiguity of the assembled genome, several metrics (e.g., centromeres, telomeres, and $N_{50}$) were evaluated. Centromeric regions were identified by the enrichment of *Copia*-like transposons and CENPA ChIP-seq data[37]. To identify telomeres, the Python script TelomereSearch.py (-m 10 -l 5000) (https://github.com/jamiemcg/TelomereSearch) was used to search for the TTTAGGG or CCCTAAA motif that has been proposed for oomycete telomeric sequences[55]. $N_{50}$ and coverage depth were calculated by QUAST v5.0.2[56]. Genome assembly completeness was measured by BUSCO v4.1.2 using the stramenopiles_odb10 database for reference[57].

### Genome annotation
To predict gene models, genome repeat sequences were masked using RepeatMasker v4.1.0 (http://repeatmasker.org), combining the Repbase v20181026 library with repeat consensus sequences generated by RepeatModeler v2.0.2[58]. Polished PacBio full-length and non-chimeric reads of the transcriptome were aligned to the reference genome using GMAP v2020-10-14[59]. Full-length transcripts were predicted using TransDecoder v5.5.0. Then, complete gene models, including 5′ and 3′ untranslated regions, were extracted as the training dataset input. Strand-specific RNA-seq reads were mapped to the genome using HISAT2 v2.2.1 (--rna-strandness RF)[60]. Reads of plus or minus strand were separated by SAMtools v1.2[61]. Subsequently, BAM files were used to extract splicing sites according to intron hint evidence by Augustus v3.4.0 (bam2hints)[62]. Finally, gene models were annotated by Augustus using de novo training species parameters and intron hint evidence. Long terminal repeat (LTR) retrotransposons were identified and annotated by LTR_retriever v2.9.0[63] and LTRharvest (GenomeTools) v1.6.2[64].

### Whole-genome synteny analysis
Whole-genome protein synteny alignment was visualized by TBtools v1.098[65] and the SynVisio website[66,67]. The Circos plot was visualized by Circos v0.69[68]. BEDTools v2.29 was used to calculate sliding window size and coverage[69]. Tandem duplicated genes were identified using the DupGen_finder pipeline, and *G. ultimum* was used as the outgroup[70].

### SV and SNP detection
SVs were detected by SyRI v1.5.4[71]. Collinear contigs of *P. sojae* 2019 were anchored to the chromosomes using SyRI (chroder). SVs located at the *P. sojae* 2019 contig breakpoint were manually removed. For SNP identification, the resequencing reads of 25 *P. sojae* isolates were mapped to the *P. sojae* 2023 genome using BWA v0.7.17[72]. The 25 *P. sojae* isolates were downloaded from NCBI (BioProject ID:

PRJNA578597). SNP calling and filtering were conducted by SAMtools v1.11 and BCFtools v1.14, using parameters (GQ>=20 DP>=5 -g 5)[40]. CNVs (copy number variations) were calculated by CNVnator v0.4.1[73].

### Gene family and effector annotation
The prediction of RxLR genes was conducted using iterative HMM searches and BLASTP searches for RxLR proteins, checking for the presence of R/Q/GxLR or EER sequences within the 21st to 120th residues of the protein sequences[74]. CRN was identified by a Hidden Markov Model (default parameters) combined with an LFLAK motif search[8]. CAZymes, elicitins, transporters, and transcription factors were annotated by searching for known homologous proteins in *P. sojae* v1.1 using BLASTP v2.9.0[75,76]. Whole-genome Pfam domains and Gene Ontology terms were identified by InterProScan v5.61[77].

### Phylogenetic tree construction and inference of ancestral karyotype
The maximum likelihood and minimum evolution algorithms were used to reconstruct the phylogenetic tree. Single-copy orthologous genes were identified using OrthoFinder v2.4.0[78] (-M msa) and Sonic-Paranoid v2.0.3[79] (-m sensitive --mmseqs 5.0 --max-len-diff 0.1 --min-bitscore 50 --complete-aln --inflation 2.5), with the intersection yielding a set of 24 single-copy genes. These 24 single-copy genes were concatenated for phylogenetic analysis. Sequence parameter calculation was based on the methods of Marco Thines et al.[80], and trees were constructed using MEGA v11[81] (Gamma distribution is 0.4093) and IQ-TREE v2.1.4[82] (-B 10000). The ancestral gene sets and gene orders for 27 oomycete species ancestors were reconstructed using AGORA v3.1[83] with iterative reconstructions and constrained gene families. The occurrence of chromosome fusions and fissions were calculated using A0 karyotype as reference. The karyotype plot was calculated and visualized by 'misc.compareGenomes.py' script.

### Structure clustering and network graph generation
GO enrichment analysis was accomplished using clusterProfiler v4.0[84]. Genes in fused regions were selected as the target genes, and performed GO enrichment analysis using the whole genome gene set as the background. Signal peptide was predicted using SignalP v3.0[85], with HMM score >0.9 as cutoff. A total of 1004 proteins with signal peptides were selected to predict structures. Structures were predicted using AlphaFold2[86] and similarity network were done using Gephi v0.10.1 (https://github.com/gephi). We calculated the TM-score between two proteins using TM-align[87] and considered them structurally similar when the bidirectional score was greater than 0.5. ChimeraX v1.6.1 was used to visualize the 3D protein structures[88].

### CRISPR/Cas9-mediated gene knockouts
We knocked out *GOS280* gene using CRISPR-mediated gene knockout methods in *P. sojae*[89]. The *hph* gene, flanked by 1.0 kb fragments on both upstream and downstream, was used as the HDR donor DNA. To screen for transformants, we employed V8 medium containing 50 μg/ml G418. Primer combinations, as shown in Supplementary Fig. 20, were used for further verification.

### Plasmid construction and agroinfiltration of *N. benthamiana*
Coding regions of candidate effectors were amplified from *P. sojae* genomic DNA. Primers were shown in Supplementary Data 6. The resulting fragments were purified and inserted into the pBIN-*p35S: GFP* vector utilizing C115 ligase. Each construct underwent verification through sequencing before being transformed into the *Agrobacterium tumefaciens* strain GV3101. The heat shock method was employed for *A. tumefaciens* transformation, and the single colonies that grew after coating were verified by colony PCR again.

*A. tumefaciens* GV3101 carrying plasmids were cultured for 16 h in LB medium supplemented with kanamycin (50 mg/L) and rifampicin

(50 mg/L) at 30 °C, 200 rpm. *A. tumefaciens* cultures were collected by centrifugation at 3,600 *g* for 5 min, washed for three times with ATTA buffer (10 mM MgCl$_2$, 10 mM MES [pH = 5.7], 100 μM acetosyringone). *A. tumefaciens* cultures were adjusted to appropriate concentrations, with OD$_{600}$ = 0.8 for candidate effectors, OD$_{600}$ = 0.4 for INF1, and OD$_{600}$ = 0.2 for P19, and were infiltrated into 4–6-week-old *N. benthamiana* leaves using clean needleless syringes. When injecting different candidate effectors, sterile syringes and gloves are replaced to avoid cross-contamination. Infiltrated *N. benthamiana* were maintained in the greenhouse for 48 h and the infiltrated leaves were collected for protein accumulation. Each candidate effector has at least nine replicates on different leaves.

## Statistics and reproducibility
Statistical analyses were conducted using R, GraphPad Prism, and Excel. All data are presented as mean ± SD. The sample sizes chosen were appropriate for the statistical analyses performed in this study. No data were excluded. The design and data collection of experiments were randomized, and infection assays were conducted blind. The results of all key experiments were reproducibly confirmed. Source data are provided as a Source Data file.

## Reporting summary
Further information on research design is available in the Nature Portfolio Reporting Summary linked to this article.

## Data availability
Supporting genome and annotation data for this study are available on Zenodo [https://doi.org/10.5281/zenodo.11098592]. The raw sequencing data generated in this study have been deposited in the NCBI BioProject database under accession code PRJNA910369. The newly assembled *P. sojae* genome in this study has been deposited in the NCBI under accession code PRJNA1106983. RNA-seq and ATAC-seq data used in this study are under BioProject accession codes PRJNA426510 and PRJNA761250, respectively. Genome assembly and GFF annotation files were downloaded from NCBI. GenBank assembly accessions are listed as follows: GCA_009848525.1, GCA_000149755.2, GCA_021491655.1, GCA_004359215.2, GCA_026184515.1, GCA_026225685.1, and GCA_030027945.1. CENPA ChIP-seq data used in this study are under BioProject accession code PRJNA563922. Source data are provided with this paper.

## Code availability
The codes are openly available at http://gitee.com/biozzc/genome_analysis.

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

## Acknowledgements

This research was supported by grants from the National Key R&D Program of China (2022YFD1400801) and the National Natural Science Foundation of China (32172374; 31721004; 31972250). Bioinformatics analyses were supported from the Bioinformatics Center at Nanjing Agricultural University.

## Author contributions

W.Y., Yuanchao Wang, S.D., and Yan Wang designed the research; J.C. and H.F. prepared the samples; Z.Z., W.Y., Y.T., L.W., and K.L. analyzed the data; X.Z. conducted the experiment; and Z.Z. and W.Y. wrote the paper.

## Competing interests

The authors declare no competing interests.
