## [Peer Review File · Nature Communications]

Complete telomere-to-telomere genomes uncover virulence evolution conferred by chromosome fusion in oomycete plant pathogensREVIEWER COMMENTS

Reviewer #1 (Remarks to the Author):

Overall, the manuscript is interesting and novel, but there are some issues as outlined below that require some major revisions and come recalculation and, if some claims regarding the ANK effector candidates should be kept, some additional experimentation.

Line 19. Globisporangium ultimum.

Line 21. Peronosporales.

Line 22. There is no compelling evidence provided for an involvement of Copia elements. Figure 3 does not seem to support this.

Line 24. pathogenicity genes.

Line 31. effector evolution.

There are many small grammar mistakes throughout the entire manuscript, so the authors are encouraged to have a native English speaker proofread their manuscript or use a professional editing service.

Line 208. This description is way too short to ensure it can be repeated. Also a list of genes tested is missing from here.

Line 383. the figures on the potential ANK effectors should be upgraded to a full figure.

Line 397. It is very suspicious, if various repeat numbers can trigger cell death. This is suggestive of an artefact / an effect of subsequent oligomerization. This should be discussed and potentially an experiment designed to account for this, e.g. the low-repeat proteins could be fused with a protein not inciting cell-death but hampering oligomerization.

I am jumping here to the figures, as there are some issues that are best pointed out on their basis.

Figure 2 needs revision. The clade with "Pythium" ultimum is Globisporangium. "Pythiales" are not recognised on the order level anymore (see also the most recent Outline of Fungi), use Peronosporales instead. Do not display a topology tree, this is misleading, as it obscures evolutionary distances. "Stramenopiles" should read "Straminipila", which is the correct spelling if used as a taxonomic and not as an informal designation. The disjunction of downy mildews is most likely and artefact of likelihood (also present in Bayesian inference) reconstruction, as likelihood tends to group species with high mutation rates with those having a low mutation rate in case of low taxon sampling in the fast-evolving group. This is actually also supported by the chromosome reconstructions in the study, as evident from the various features shared by the two downy mildews - the small "cyan" chromosome, the small grey/cyan chromosome, the small salmon coloured chromosome, and the very similar light green/grey/brown chromosome. The latter is also there (even though a bit bigger in *Phytophthora infestans*, which is closely related to the downy mildews). As all conclusions rely on a correct topology of the tree, the authors are encouraged to use either a much higher taxon sampling in downy mildews, or to use a phylogenetic method without such artifacts, e.g. minimum evolution. Alternatively, they could calculate the tree topology based on chromosome similarity, which would, apparently also lead to the correct inference. In addition, it is important that b and c use the same set of species (which should be as big as possible).

Figure 3 does not support that Copia elements are involved in the Fusion/Fission events.

Reviewer #2 (Remarks to the Author):

The manuscript 'Complete telomere-to-telomere genomes uncover virulence evolution conferred by chromosome fusion in oomycete plant pathogens' by Zhang and colleagues reports on the telomere-to-telomere genome assemblies of the oomycete plant pathogens *Phytophthora sojae*, *Pythium ultimum*, *Pythium oligandrum*, and *Pythium spinosum*. These genomic resources will be extremely valuable for the research community working on these important plant pathogens.

The manuscript reports on comprehensive and high-quality comparative genomic analyses to study chromosomal structures in several oomycetes and to link large-scale chromosomal variation to the evolution of virulence. While the here presented work provides solid evidence that chromosomal fusions and fissions are important in shaping the karyotypes of extent oomycetes (e.g., L20 or Fig. 2), frequent chromosomal fusion had been previously reported by Fletcher and colleagues (2022; 2023); this contrasts with the manuscript that insinuated that chromosome-level comparisons have not been previously performed (L84). Moreover, the number of ancestral chromosomes ($n=17$) had been similarly proposed previously. Thus, the observation of frequent occurrence of karyotype variation in oomycetes caused by fusion/fission is per se not novel, which needs to be much clearer acknowledged and especially the discussion needs to be significantly toned down (e.g., L442). Nevertheless, the in-depth study of dynamic genomic regions formed by chromosomal fusion is interesting for researchers in the field, as it provides insights into mechanisms that contribute to effector evolution.

The manuscript mentions that specific transposon families (Copia elements) are enriched at fusion sites and that these might contribute to chromosomal fusions. Based on the distances of the fusions to these elements visible in fig. 3, this enrichment does not seem to be very pronounced. To substantiate this notion, specific occurrences of these elements at fusion sights (or their distance) need to be further quantified. Moreover, it is mentioned that there are differences between insertion times and type of transposon in fused and non-fused chromosomes (fig 4c), yet these, especially for Gypsy, look very much alike. Are these differences statistical supported? The manuscript also shows that duplicated effectors occur in proximity of telomeres, especially in fused chromosomes. Could the authors speculate how mechanistically duplications occur more frequently at fused chromosomes?

The manuscript mentions several times that a novel approach to identify effectors has been developed (e.g., L29). First, it is not clearly described how evolutionary fused regions (EFR) are defined? Are these regions as suggested (L323) indeed more variable (e.g., higher frequency of SNPs or SVs) than the remainder of the genome, or the sub-telomeric regions? It is particularly unclear how the ERF relates to the rearranged genomic regions that are subsequently mined for novel effectors (L348ff). Second, exploiting the identification of dynamic regions between species/strains to aid effector identification is not novel, and has been used to identify numerous (a)virulence effectors in fungal and oomycete pathogens. Third, clustering of protein structures to reveal families with shared or novel folds has recently performed for plant pathogens, including oomycetes (Derbyshire et al. 2023; Seong et al. 2023), and thus is not novel. Arguably, the ankyrin repeat families could be easily identified as likely relevant based on their abundance alone (see also comments below), and structural predictions would have not necessary. While interesting cases where RXLR effectors could be matched to those with known function are shown, the provided evidence is collectively not too compelling and thus I am not too convinced that the here mentioned 'novel approach' is instrumental to identify novel and relevant effector candidates.

The manuscript reports on high-quality genome assemblies, largely robust analyses with appropriate approaches/tools, and clear visual representations. However, I also would like to note that the manuscript contains many spelling and grammatical mistakes (see some examples below). Moreover, some sentences are not entirely clear and/or correct and need to be critically revised. For example, in lines 39, the concepts of effectors and of co-evolutionary arms races are introduced. However, 'pathogenic secreted proteins' do not exist; proteins can be secreted, and proteins can contribute to virulence/pathogenicity/establishment of infection, none of which is captured by the phrase used here. Similarly, effectors do not have an 'evolutionary mechanisms' (L45) that can be investigate or are 'on the basis of genome architecture' (L47). Rather, there are evolutionary mechanisms that can be investigated, and genome architecture and effector evolution

are interlinked. These are only few examples, but these concerns, throughout the manuscript, need to be addressed prior to acceptance.

Additional comments and suggestions:

L37: 'Phylogenetic analyses revealed that the plant pathogenic oomycetes independently...' Based on the data shown in Figure 2, this statement is not correct. Plant pathogenetic oomycetes are not divided into three lineages, especially since *Pythium* are not monophyletic, as are the downy mildew. Thus, the phylogenetic associations within oomycete need to be correctly introduced.

L54: 'In addition to the 'two-speed genome' model, several effector (including Avr) genes adjacent to telomere repeats and/or arranged in a clustered distribution were identified in a few high-quality fungal genomes^{12,13,14}.' I do not understand why this seems to 'in addition to the two-speed model'. The model conceptualizes genome evolution in filamentous plant pathogens (oomycetes and fungi), and the observed clustering of effectors in repeat-rich regions in some fungi, as mentioned here, just perfectly fit the proposed conceptual model. If there is any real 'addition' to the model, this needs to be much better explained.

L84: 'lack'  'lacking'

L91: 'fusion, which confer the novel vitality of effector evolution.' I am not sure what 'confer the novel vitality' is supposed to mean. Could the authors please rephrase this sentence?

L234: 'obviously'  delete this word

L243: it is mentioned that DNA repeats occur frequently in proximity to RXLR effector. This seems to contrast the introduction where LTRs have been mentioned to be implied into driving the evolution of the two-speed genome, and effector genes. Can the authors attempt to reconcile these statements? What could explain these differences?

L281: What is the difference between 'a dynamic and plastic process'? either these terms are redundant or need to be explained

L348: The number of predicted genes seems to be large compared with other oomycetes. Are these numbers inflated since many transposons are erroneously annotated as genes, and thus at least partially explain why so many genes (~40%) occur at rearranged regions?

L369: overrepresentation of ANK repeat proteins in plant pathogen, especially in oomycetes, has been previously reported by Seidl and colleagues (2012).

Fig. S5: Genes in relationship to transposable elements (e.g., reverse transcriptase, integrase, or retrotransposons) are enriched in duplications. This enrichment is not surprising and indicates that a large proportion of the reported duplication are in fact just transposons, and thus the reported number of duplicates as well as the gene annotations need to be cleaned of these 'genes'

We thank the editor and reviewers for their thoughtful and detailed input. We have done our best to address all the comments. Our detailed responses are as follows.

REVIEWER COMMENTS

Reviewer #1 (Remarks to the Author):

Overall, the manuscript is interesting and novel, but there are some issues as outlined below that require some major revisions and come recalculation and, if some claims regarding the ANK effector candidates should be kept, some additional experimentation.

Thank you very much for your recognition. Thank you for your valuable comments. We have responded to them point by point, and these modifications have helped us improve significantly.

Q1: Comment: Line 19. *Globisporangium ultimum*.

R1: Corrected. '*Pythium ultimum*' and '*Pythium spinosum*' have been updated to '*Globisporangium ultimum*' and '*Globisporangium spinosum*'.

Q2: Line 21. Peronosporales.

R2: Corrected.

Q3: Line 22. There is no compelling evidence provided for an involvement of *Copia* elements. Figure 3 does not seem to support this.

R3: In the revised version, we updated **Fig. 3**. Further quantitative analysis was performed on the distance of *Copia* elements to fusion/fission regions, with corresponding marking of centromere positions. This more clearly illustrates the relationship between the *Copia*-enriched regions and fusion/fission events.

Q4: Line 24. pathogenicity genes.

R4: Corrected.

Q5: Line 31. effector evolution.

R5: Corrected.

Q6: There are many small grammar mistakes throughout the entire manuscript, so the authors are encouraged to have a native English speaker proofread their manuscript or use a professional editing service.

R6: Thanks for your suggestions. The entire text has been carefully revised and checked by a native English speaker.

Q7: Line 208. This description is way too short to ensure it can be repeated. Also a list of genes tested is missing from here.

R7: The detailed description and sequence information have been added. A list of genes is shown in **Supplementary Table 6**.

Q8: Line 383. the figures on the potential ANK effectors should be upgraded to a full figure.

R8: Thanks. We have upgraded the potential ANK effectors and several functional validations

to the new **Fig. 6**.

Q9: Line 397. It is very suspicious, if various repeat numbers can trigger cell death. This is suggestive of an artefact / an effect of subsequent oligomerization. This should be discussed and potentially an experiment designed to account for this, e.g. the low-repeat proteins could be fused with a protein not inciting cell-death but hampering oligomerization.

R9: Thanks for your suggestions. Oligomerization of secreted ANK proteins is an interesting idea. In next work, we will further conduct in-depth functional studies on them, including the deletion of repetitive domains, the attempt to fuse 'helper' proteins, and the establishment of an experimental model for interfering with oligomerization through structural prediction and experiments. It is instructive for the subsequent in-depth study of the function of this family.

I am jumping here to the figures, as there are some issues that are best pointed out on their basis.

Q10: Figure 2 needs revision. The clade with "Pythium" ultimum is Globisoirangium. "Pythiales" are not recognised on the order level anymore (see also the most recent Outline of Fungi), use Peronosporales instead. Do not display a topology tree, this is misleading, as it obscures evolutionary distances. "Stramenopiles" should read "Straminipila", which is the correct spelling if used as a taxonomic and not as an informal designation. The disjunction of downy mildews is most likely an artefact of likelihood (also present in Bayesian inference) reconstruction, as likelihood tends to group species with high mutation rates with those having a low mutation rate in case of low taxon sampling in the fast-evolving group. This is actually also supported by the chromosome reconstructions in the study, as evident from the various features shared by the two downy mildews - the small "cyan" chromosome, the small grey/cyan chromosome, the small salmon coloured chromosome, and the very similar light green/grey/brown chromosome. The latter is also there (even though a bit bigger) in *Phytophthora infestans*, which is closely related to the downy mildews. As all conclusions rely on a correct topology of the tree, the authors are encouraged to use either a much higher taxon sampling in downy mildews, or to use a phylogenetic method without such artifacts, e.g. minimum evolution. Alternatively, they could calculate the tree topology based on chromosome similarity, which would, apparently also lead to the correct inference.

R10: These professional suggestions are very helpful! We have sampled additional downy mildew species (4 to 12), excluded lineages unrelated to this study, and reconstructed the species phylogeny. Using both maximum likelihood and minimum evolution methods, refer to a recent study (Thines M et al., 2023), we constructed a new consensus tree, supporting the monophyly of downy mildews. Please see the following figure.

Q11: In addition, it is important that b and c use the same set of species (which should be as big as possible).

R11: So far, we have collected the chromosome-level genomes of previously published oomycetes as much as possible. To ensure the acquisition of contiguous ancestral regions, the revised results are presented at the chromosomal level for more species (4 to 6). The results of collinearity and reconstruction are highly consistent. More details are presented in the new Fig. 2b, c.

The reference list is below:

- (1) Thines, M., Mishra, B. & Ploch, S. Multigene analyses with a broad sampling in *Phytophthora* and related genera provide evidence for the monophyly of downy mildews. *Mycol Progress* 22, 82 (2023).

Q12: Figure 3 does not support that Copia elements are involved in the Fusion/Fission events.

R12: Corrected. We have revised the corresponding results, as stated in R3. For your convenience, the results are displayed below.

Reviewer #2 (Remarks to the Author):

The manuscript ‘Complete telomere-to-telomere genomes uncover virulence evolution conferred by chromosome fusion in oomycete plant pathogens’ by Zhang and colleagues reports on the telomere-to-telomere genome assemblies of the oomycete plant pathogens *Phytophthora sojae*, *Pythium ultimum*, *Pythium oligandrum*, and *Pythium spinosum*. These genomic resources will be extremely valuable for the research community working on these important plant pathogens.

Thank you for recognizing our work.

Q1: The manuscript reports on comprehensive and high-quality comparative genomic analyses to study chromosomal structures in several oomycetes and to link large-scale chromosomal variation to the evolution of virulence. While the here presented work provides solid evidence that chromosomal fusions and fissions are important in shaping the karyotypes of extant oomycetes (e.g., L20 or Fig. 2), frequent chromosomal fusion had been previously reported by Fletcher and colleagues (2022; 2023); this contrasts with the manuscript that insinuated that chromosome-level comparisons have not been previously performed (L84).

RI: Thanks for pointing out these questions. In previous work, Kyle Fletcher et al. pointed out “It remains unresolved whether these are true chromosomal fusions or assembly errors in *Phytophthora sojae*. Comparison with a more recent long-read assembly of *P. sojae* did not resolve this because the assembly was too fragmented (Fletcher K et al., 2022).” In their subsequent work, they only compared the fusion events occurring between two species, *Per. sorghi* and *Pe. effusa*. “The lineage leading to *P. sorghi* had undergone four chromosome fusions since diverging from the last common ancestor with *P. effusa*.” (Fletcher K et al., 2023).

We have introduced the findings of Fletcher et al (L59-60). As the reviewer said, by integrating genomic data from *Phytophthora*, downy mildew, and *Pythium*, we substantiate the widespread occurrence of chromosomal fusion in oomycetes. According to our analysis, we found that the proportion of chromosomes with complete telomeres in previously published articles is as follows: 17/17 for *Pe. effusa*, 0/19 for *B. lactucae*, 1/13 for *Per. sorghi*, 0/15 for *P. infestans*, and 6/18 for *P. plurivora*. Thus, T2T-level genome assemblies are lacking.

We acknowledge that our emphasis on ‘largely lacking’ and ‘deep comparative’ analysis may be not precise. We have modified the statements to ‘T2T-level genome assemblies are still limited, and a comprehensive chromosome-level comparative genomic analysis in oomycetes is still insufficient.’ and incorporated findings from Fletcher et al. in the introduction to improve the precision of our statements.

The reference list is below:

- (1) Fletcher K, Martin F, Isakeit T, Cavanaugh K, Magill C, Michelmore R. The genome of the oomycete *Peronosclerospora sorghi*, a cosmopolitan pathogen of maize and sorghum, is inflated with dispersed pseudogenes. *G3 (Bethesda)*. 2023;13(3).

- (2) Fletcher K, Shin OH, Clark KJ, et al. Ancestral chromosomes for family Peronosporaceae inferred from a Telomere-to-Telomere genome assembly of *Peronospora effusa*. *Mol Plant Microbe Interact.* 2022;35(6):450-463.

Q2: Moreover, the number of ancestral chromosomes (n=17) had been similarly proposed previously. Thus, the observation of frequent occurrence of karyotype variation in oomycetes caused by fusion/fission is per se not novel, which needs to be much clearer acknowledged and especially the discussion needs to be significantly toned down (e.g., L442).

R2: Thanks for pointing out these questions. Karyotype variation has been reported between two species, *Per. sorghi* and *Pe. effusa* (Fletcher K et al., 2023). We must acknowledge that in L442, our statement is incorrect. We have modified the corresponding statement to ‘we generated multiple T2T genomes assemblies for oomycetes and demonstrated the widespread chromosome fusion and fission phenomenon in oomycetes, which further expanded the findings of Fletcher K et al.’. This question ‘the number of ancestral chromosomes (n=17) had been similarly proposed previously ...’ has also aroused our confusion and attention during the research. The conclusion that there are 17 ancestral chromosomes may arise from the statement in the original text: ‘The most recent common ancestor of *Peronospora effusa* and *Phytophthora sojae* (*Phytophthora* clade 7) is more ancient than the common ancestor of *Peronospora effusa* and *B. lactucae*. Given the high levels of synteny between these three species (Fig. 5A and B), it is likely that the 17-chromosome architecture of *P. effusa* and *B. lactucae* will be ancestral to hundreds of other species in the Peronosporaceae family.’ (Fletcher K et al., 2022). *P. sojae* has 12 chromosomes, and *Pe. effusa* has 17 chromosomes. The difference in chromosome numbers between the two species indicates that relying solely on the synteny of the three species may be insufficient to conclude that the ancestor had a 17-chromosome architecture, especially when *P. sojae* had fragmented assemblies.

In this study, we conducted ancestral karyotype reconstruction, a well-established method in mammalian chromosomal evolution analysis (Muffato M et al., 2023). *Pe. effusa* has 17 chromosomes, coincidentally matching the reconstructed ancestral chromosome number. Interestingly, most chromosomes of *Pe. effusa* matched to the ancestral karyotype. We have modified the corresponding statements and discussed the 17 ancestral chromosomes proposed in previous studies. Modified phrase was ‘Similar to the conclusion of previous study (Fletcher K et al., 2022), reconstructed ancestral karyotype also proposed that there may be 17 ancestral chromosomes, which is close to the number of 18 conserved chromosomes in Pythiaceae.’

Muffato M et al. *Nat Ecol Evol.* 2023;7(3):355-366.

The reference list is below:

- (1) Muffato M, Louis A, Nguyen NTT, Lucas J, Berthelot C, Roest Crolius H. Reconstruction of hundreds of reference ancestral genomes across the eukaryotic kingdom. *Nat Ecol Evol.* 2023;7(3):355-366.

Q3: Nevertheless, the in-depth study of dynamic genomic regions formed by chromosomal fusion is interesting for researchers in the field, as it provides insights into mechanisms that contribute to effector evolution.

R3: Your idea is very valuable! As the current data are limited, we plan to conduct chromosome-level assembly on additional *P. sojiae* isolates in the future. We aim to integrate intraspecific and interspecific comparative genomics, epigenetics, three-dimensional chromatin interaction, and open chromatin sequencing to conduct comprehensive research in this area.

Q4: The manuscript mentions that specific transposon families (*Copia* elements) are enriched at fusion sites and that these might contribute to chromosomal fusions. Based on the distances of the fusions to these elements visible in fig. 3, this enrichment does not seem to be very pronounced. To substantiate this notion, specific occurrences of these elements at fusion sites (or their distance) need to be further quantified.

R4: Great suggestion! We magnified and quantified the regions of chromosome fusion/fission, revealing a clear enrichment and loss of *Copia* transposons in these areas. Further details are presented in the revised manuscript (Fig. 3, Supplementary Fig. 7, and Supplementary Fig. 8). For your convenience, the results of Fig. 3 are displayed below.

Q5: Moreover, it is mentioned that there are differences between insertion times and type of transposon in fused and non-fused chromosomes (fig 4c), yet these, especially for Gypsy, look very much alike. Are these differences statistical supported?

R5: Thank you for your insights. *Gypsy* transposons are indeed alike between fused and non-fused chromosomes. We found that there was no significant difference in the density of *Copia* and *Gypsy* between fused and non-fused chromosomes (**Supplementary Fig. 9**). Interestingly, in fused chromosomes, *Copia* insertion time appear to be more recent compared to non-fused chromosomes. Considering the essential role of *Copia* in centromeres, we speculate that the process of chromosome fusion might have triggered a reshaping of centromeres, leading to the observed *Copia* dynamics. This may be associated with chromosome fusion and fission driven by centromeres. For your convenience, the revised results of **Fig. 4c** are displayed below.

Q6: The manuscript also shows that duplicated effectors occur in proximity of telomeres, especially in fused chromosomes. Could the authors speculate how mechanistically duplications occur more frequently at fused chromosomes?

R6: By reviewing several literatures, we found interesting correlations between subtelomeric duplications and transposons. The subtelomeric regions play a crucial role as hotspots for effector duplication and rapid evolution in *Magnaporthe oryzae* and *Colletotrichum*. Dynamic segmental duplications often occur between the core chromosome ends and mini-chromosomes (Orbach et al., 2000; Chuma et al., 2011; Pamela G et al., 2021). The mechanism underlying the interchange between core chromosomes and mini-chromosomes is not yet fully understood. However, the ends of core chromosome and mini-chromosomes are both exhibit enrichment of DNA transposons and LINE retrotransposons, suggesting a potential involvement of non-allelic homologous transposon-mediated recombination (Orbach et al., 2000; Peng Z et al., 2019).

Based on the above findings, we speculated that this mechanism may also exist in *P. sojae*. we observed regions where sub-telomere replication occurs, and it is intriguing that these regions are indeed rich in DNA transposons, suggesting that DNA transposons may mediate the segmental duplication of chromosomes, especially in the fused chromosome Chr02 and Chr03 (Supplementary Fig. 11). We also noted that other inter-chromosome duplicated regions did not appear to apparent DNA transposons enrichment, implying that other mechanisms or unknown transposon elements may be involved.

Peng Z et al. *PLoS Genet.* 2019;15(9):e1008272.

The reference list is below:

- (1) Orbach MJ, Farrall L, Sweigard JA, Chumley FG, Valent B. A telomeric avirulence gene determines efficacy for the rice blast resistance gene Pi-ta. *Plant Cell.* 2000;12(11):2019-2032.
- (2) Chuma I, Isobe C, Hotta Y, et al. Multiple translocation of the AVR-Pita effector gene among chromosomes of the rice blast fungus *Magnaporthe oryzae* and related species. *PLoS Pathog.* 2011;7(7): e1002147.
- (3) Gan P, Hiroyama R, Tsushima A, et al. Telomeres and a repeat-rich chromosome encode effector gene clusters in plant pathogenic *Colletotrichum* fungi. *Environ Microbiol.* 2021;23(10):6004-6018.
- (4) Peng Z, Oliveira-Garcia E, Lin G, et al. Effector gene reshuffling involves dispensable mini-chromosomes in the wheat blast fungus. *PLoS Genet.* 2019;15(9): e1008272.

Q7: The manuscript mentions several times that a novel approach to identify effectors has been developed (e.g., L29). First, it is not clearly described how evolutionary fused regions (EFR) are defined?

R7: For consistency with previous studies, we renamed it evolutionary breakage regions (EBRs) (Kim J et al., 2017; Fan H et al., 2019), which we defined in **Supplementary Fig. 14**.

The reference list is below:

- (1) Kim J, Farré M, et al. Reconstruction and evolutionary history of eutherian chromosomes. *Proc Natl Acad Sci U S A.* 2017;114(27): E5379-E5388.
- (2) Fan H, Wu Q, Wei F, Yang F, Ng BL, Hu Y. Chromosome-level genome assembly for giant panda provides novel insights into Carnivora chromosome evolution. *Genome Biol.* 2019;20(1): 267.

Q8: Are these regions as suggested (L323) indeed more variable (e.g., higher frequency of SNPs or SVs) than the remainder of the genome, or the sub-telomeric regions?

R8: As mentioned earlier, this is a good idea! These regions are species-specific, breakpoint regions between species. However, there was no significant difference in genomic variations such as SNPs and CNVs among 24 different isolates, although there are still many variants

(Supplementary Fig. 14). We speculate that these regions may not select a higher mutation rate for the stability of the genome after speciation. The next-generation sequencing data may still have limitations for in-depth analysis of these regions, and the collinearity comparison at the chromosome level of multiple isolates will be very informative. In the future, more isolates genomic data at the chromosome level will help us to further explore new findings. For example, are there differences in epigenetic modification in these regions? This may require extensive additional analysis to further determine. Thank you. This is a very good point.

Q9: It is particularly unclear how the ERF relates to the rearranged genomic regions that are subsequently mined for novel effectors (L348ff).

R9: We mapped reconstructed ancestral chromosomes to the current *P. sojae* genome, and all non-collinear regions contained chromosomal breakpoints and gene rearrangement regions. These regions were used to mine candidate effectors. The definition was show in Supplementary Fig. 14.

Q10: Second, exploiting the identification of dynamic regions between species/strains to aid effector identification is not novel, and has been used to identify numerous (a)virulence effectors in fungal and oomycete pathogens. Third, clustering of protein structures to reveal

families with shared or novel folds has recently performed for plant pathogens, including oomycetes (Derbyshire et al. 2023; Seong et al. 2023), and thus is not novel.

R10: We appreciate the reviewer for insightfully finding this issue. Our previous wording was indeed inappropriate, and we have made the necessary revisions. We no longer emphasize that it is ‘novel’, but analyze the effector by integrating the above two classic methods.

Identifying effectors through dynamic regions is a classic and well-known method, these regions are undergoing swift evolution, and usually show significant sequence polymorphism (Jiang RH et al., 2008). Structure prediction is a powerful tool for clustering effectors with sequence-unrelated but structure-related features (Seong et al., 2023), but its computational process is time-consuming. In theory, the combination of the two methods is expected to facilitate the identification of candidate families.

Derbyshire et al. did not incorporate oomycetes in their analysis, and Seong et al. only introduced *Phytophthora infestans* as an outgroup, as shown in the figure below.

Seong K, Krasileva KV. *Nat Microbiol.* 2023;8(1):174-187.

The reference list is below:

- (1) Derbyshire MC, Raffaele S. Surface frustration re-patterning underlies the structural landscape and evolvability of fungal orphan candidate effectors. *Nat Commun.* 2023;14(1):5244.
- (2) Jiang RH, Tripathy S, Govers F, Tyler BM. RXLR effector reservoir in two *Phytophthora* species is dominated by a single rapidly evolving superfamily with more than 700 members. *Proc Natl Acad Sci U S A.* 2008;105(12):4874-4879.
- (3) Seong K, Krasileva KV. Prediction of effector protein structures from fungal phytopathogens enables evolutionary analyses. *Nat Microbiol.* 2023;8(1):174-187.

Q11: Arguably, the ankyrin repeat families could be easily identified as likely relevant based on their abundance alone (see also comments below), and structural predictions would have not necessary. While interesting cases where RXLR effectors could be matched to those with known function are shown, the provided evidence is collectively not too compelling and thus I am not too convinced that the here mentioned ‘novel approach’ is instrumental to identify novel and relevant effector candidates.

R11: Ankyrin-repeat family is relatively abundant in *Saprolegnia parasitica* (Torto-Alalibo T et al., 2005), and is the largest family in *Pythium ultimum* (Lévesque CA et al., 2010). Ankyrin genes also show up-regulation in germinating cysts in *S. parasitica*, which suggests the importance in the initial stages of host colonization (Jiang RH et al., 2013). Our study reinforces previous findings, suggesting the significance of the ANK family in oomycetes.

Given the vastness of this family, its members exhibit considerable sequence and structural diversity. Meanwhile, research on secreted ANK is relatively limited. They are more likely to be sequence-unrelated but structurally similar, referring to the study by Seong K et al. on the evolution of the RNase effector family (Seong et al., 2023). We supplemented these results in **Supplementary Fig. 17**. Therefore, structure prediction and clustering remains crucial to digest ANK family for a more in-depth investigation. By visualizing the structures of different ANKs, we could find several interesting folding types, such as “donuts”. **Supplementary Fig. 17** is shown below.

Our research further advances the work related to ANKs. We conducted a preliminary exploration of the evolution and function of secreted ANKs. In the discussion, we pointed out more clearly the findings of Seidl et al., and discussed our progress.

At present, the method has not been tested by more genomic data, our statement indeed is not appropriate and has been modified in the revised manuscript. In the future, we can consider using this combination method to make some new attempts on the genome of *Pythium* or several fungi.

Jiang RH et al. *PLoS Genet.* 2013;9(6):e1003272.

Table 1 Most represented protein domains in the *Saprolegnia parasitica* EST contigs

From: Expressed sequence tags from the oomycete fish pathogen *Saprolegnia parasitica* reveal putative virulence factors

InterPro ID	Description	Number of contigs
IPR000719	Catalytic protein kinase	22
IPR004006	Actin and actin-like	11
IPR001993	Mitochondrial energy transfer proteins (various)	9
IPR000676	Ustilagin domain	6
IPR002110	Ankyrin repeat	5
IPR011430	Cysteine-rich WY-VE repeats	5
IPR001795	GTP-binding elongation factor	5
IPR011353	Multispecific proteasome peptidase	5

Torto-Alalibo T et al. *BMC Microbiol.* 2005;5:46.

The reference list is below:

- (1) Torto-Alalibo T, Tian M, Gajendran K, Waugh ME, van West P, Kamoun S. Expressed sequence tags from the oomycete fish pathogen *Saprolegnia parasitica* reveal putative virulence factors. *BMC Microbiol.* 2005; 5:46.
- (2) Lévesque CA, Brouwer H, Cano L, et al. Genome sequence of the necrotrophic plant pathogen *Pythium ultimum* reveals original pathogenicity mechanisms and effector repertoire. *Genome Biol.* 2010;11(7): R73.
- (3) Jiang RH, de Bruijn I, Haas BJ, et al. Distinctive expansion of potential virulence genes in the genome of the oomycete fish pathogen *Saprolegnia parasitica*. *PLoS Genet.* 2013;9(6): e1003272.

Q12: The manuscript reports on high-quality genome assemblies, largely robust analyses with appropriate approaches/tools, and clear visual representations. However, I also would like to note that the manuscript contains many spelling and grammatical mistakes (see some examples below). Moreover, some sentences are not entirely clear and/or correct and need to be critically revised. For example, in lines 39, the concepts of effectors and of co-evolutionary arms races are introduced. However, ‘pathogenic secreted proteins’ do not exist; proteins can be secreted, and proteins can contribute to virulence/pathogenicity/establishment of infection, none of which is captured by the phrase used here.

R12: Thank you for your expertise. We have modified the phrase to ‘Effectors are key virulence factors or toxins that alter host physiology, contributing to the establishment of infection.’

Q13: Similarly, effectors do not have an ‘evolutionary mechanisms’ (L45) that can be investigate or are ‘on the basis of genome architecture’ (L47). Rather, there are evolutionary mechanisms that can be investigated, and genome architecture and effector evolution are interlinked. These are only few examples, but these concerns, throughout the manuscript, need to be addressed prior to acceptance.

R13: Thanks. We have modified the phrase to ‘Effector evolution is tightly linked to genomic compartments.’

Additional comments and suggestions:

Q14: L37: ‘Phylogenetic analyses revealed that the plant pathogenic oomycetes independently...’ Based on the data shown in Figure 2, this statement is not correct. Plant pathogenic oomycetes are not divided into three lineages, especially since *Pythium* are not monophyletic, as are the downy mildew. Thus, the phylogenetic associations within oomycete need to be correctly introduced.

R14: Thanks. We have modified the phrase. ‘The well-known pathogens—namely, *Pythium*, downy mildews, and *Phytophthora*—presumably share a common origin, causing a lot of economic losses’

Q15: L54: ‘In addition to the ‘two-speed genome’ model, several effectors (including Avr) genes adjacent to telomere repeats and/or arranged in a clustered distribution were identified in a few high-quality fungal genomes^{12,13,14}.’ I do not understand why this seems to ‘in addition

to the two-speed model'. The model conceptualizes genome evolution in filamentous plant pathogens (oomycetes and fungi), and the observed clustering of effectors in repeat-rich regions in some fungi, as mentioned here, just perfectly fit the proposed conceptual model. If there is any real 'addition' to the model, this needs to be much better explained.

R15: Corrected. The statement is redundant.

Q16: L84: 'lack'  'lacking'

R16: Corrected.

Q17: L91: 'fusion, which confer the novel vitality of effector evolution.' I am not sure what 'confer the novel vitality' is supposed to mean. Could the authors please rephrase this sentence?

R17: Corrected. We have modified the phrase to 'which may contribute to the understanding of effector evolution.'

Q18: L234: 'obviously'  delete this word

R18: Corrected.

Q19: L243: it is mentioned that DNA repeats occur frequently in proximity to RXLR effector. This seems to contrast the introduction where LTRs have been mentioned to be implied into driving the evolution of the two-speed genome, and effector genes. Can the authors attempt to reconcile these statements? What could explain these differences?

R19: We traced the original *Nature* article. The authors proposed that LTR/*Gypsy* have a majority in the longest intergenic region. In fact, it can also be observed that DNA transposons contribute more heavily to the genomic content of regions below 3 kb (Haas BJ et al., 2009) (see below, **Fig. A**). The genome of *P. sojae* is more compact than that of *P. infestans* (**Fig. B, E**). In *P. sojae*, the distribution of the effector is denser and proportionally higher within 3 kb (Wang Y et al., 2018) (**Fig. C**). 19% of RxLRs are outside 3 kb, whereas 39% of RxLRs are within 3 kb (**Fig. D**).

LTR expansion contribute to bigger genome and sparser gene distribution in specific regions (Dong S et al., 2015) (**Fig. E**). DNA transposons are also involved in gene evolution, e.g., gene tandem duplications as well as segmental duplications (Tan S et al., 2021; Peng Z et al., 2019). We speculate that DNA repeats and LTRs may play a synergistic role in effector evolution. Simone Fouché et al. proposed that encoding key virulence factors near TEs would be a devil's bargain for pathogens. LTR expansion drives effector hopping, causing drastic changes in the genome, and plant pathogens may face long-term consequences for short-term benefits (Fouché S et al., 2022). DNA repeats might safeguard the localized, relatively modest evolution of effectors.

Editorial Note: Parts of the figure below have been redacted as indicated to remove third-party material where no permission to publish could be obtained.

A

[REDACTED]

B

[REDACTED]

C

Haas BJ et al. *Nature*. 2009;461(7262):393-398.

[REDACTED]

Wang Y et al., *Mol Plant Pathol*. 2018;19(9):2177-2186.

D

Haas BJ et al. *Nature*. 2009;461(7262):393-398.

E

- (1) Haas BJ, Kamoun S, Zody MC, et al. Genome sequence and analysis of the Irish potato famine pathogen *Phytophthora infestans*. *Nature*. 2009;461(7262):393-398.
- (2) Wang Y, Ye W, Wang Y. Genome-wide identification of long non-coding RNAs suggests a potential association with effector gene transcription in *Phytophthora sojae*. *Mol Plant Pathol*. 2018;19(9):2177-2186.
- (3) Dong S, Raffaele S, Kamoun S. The two-speed genomes of filamentous pathogens: waltz with plants. *Curr Opin Genet Dev*. 2015; 35:57-65.
- (4) Tan S, Ma H, Wang J, et al. DNA transposons mediate duplications via transposition-independent and -dependent mechanisms in metazoans. *Nat Commun*. 2021;12(1):4280.
- (5) Peng Z, Oliveira-Garcia E, Lin G, et al. Effector gene reshuffling involves dispensable mini-chromosomes in the wheat blast fungus. *PLoS Genet*. 2019;15(9): e1008272.
- (6) Fouché S, Oggenfuss U, Chanclud E, Croll D. A devil's bargain with transposable elements in plant pathogens. *Trends Genet*. 2022;38(3):222-230.

Q20: L281: What is the difference between ‘a dynamic and plastic process’? either these terms are redundant or need to be explained

R20: Corrected. These terms are redundant.

Q21: L348: The number of predicted genes seems to be large compared with other oomycetes. Are these numbers inflated since many transposons are erroneously annotated as genes, and thus at least partially explain why so many genes (~40%) occur at rearranged regions?

R21: We checked the gene annotations. A total of 1841 protein-coding genes homologous to known transposable elements were marked in the new gene annotation and excluded from the analysis. Only ~30% of the genes actually appeared in the rearranged regions.

Q22: L369: overrepresentation of ANK repeat proteins in plant pathogen, especially in oomycetes, has been previously reported by Seidl and colleagues (2012).

R22: Yes. Please see above response **R10**.

Q23: Fig. S5: Genes in relationship to transposable elements (e.g., reverse transcriptase, integrase, or retrotransposons) are enriched in duplications. This enrichment is not surprising and indicates that a large proportion of the reported duplication are in fact just transposons, and thus the reported number of duplicates as well as the gene annotations need to be cleaned of these ‘genes’

R23: Thanks for your suggestions. We have removed the transposable elements and modified the corresponding raw **Supplementary Fig. 1** to new **Supplementary Fig. 1**, and raw **Supplementary Fig. 5** to new **Supplementary Fig. 10**.

Once again, we are grateful to the editor and two professional reviewers for their valuable suggestions, which have significantly improved our manuscript.

REVIEWER COMMENTS

Reviewer #1 (Remarks to the Author):

Dear authors,

Thanks for your indepth revisions. I am happy I could help improving the manuscript.

Best regards,

Reviewer #1 aka Marco Thines

Reviewer #2 (Remarks to the Author):

The revised manuscript by Zhang and colleagues successfully addresses most of my concerns raised by the initial submission. I believe that the submission has greatly improved in terms of robustness and clarity, especially acknowledging previous work. The submitted manuscript reports on comprehensive and high-quality comparative genomic analyses to study chromosomal structures in several oomycetes and to link large-scale chromosomal variation to the evolution of virulence - I anticipate that the study will be of high relevance for scientist working on oomycetes and other plant pathogens, as well as scientist interested in genome evolution in general.

I however have a few remaining concerns/comments that have arisen from the revisions that necessitates the authors' attention.

First, based on a very valid comments raised by reviewer 1 (Q10), the manuscript now reports a novel phylogenetic analyses (Fig 2a). It is important to note that the relationship of downey mildews within Peronosporales is a sensitive topic in the research community with different and often conflicting data being reported over the years. While this is not the main topic of this article, the relationships between species are relevant for reconstructing ancestral genomes and trace evolution. While the here reported monophyly of downy mildews, based on few marker genes, seems plausible, the authors have data at hand to further corroborate these by analysing a larger set of genes (e.g., all single-copy orthologs) with alternative methods such as ML and ME. Moreover, the authors need to directly assess the impact of alternative tree topologies to the ancestral chromosome reconstruction and the number of evolutionary events necessary to reconcile the karyotypes and their evolution, given a specific phylogeny. This would not only provide a robust assessment of the karyotype evolution but might also provide additional data towards the ongoing discussion on the relationships of downey mildews.

Second, the manuscript reports on the centromeric regions. The manuscripts notes that '...', and the excessive centromeres on the fused chromosomes become inactivated. This was manifested by the absence of Copia-like sequences in these regions (Supplementary Fig. 8). '. The inactivation is not experimentally shown, and the potential relevance of Copia elements to mark potential centromeres has not been introduced. To support this firm statement, the authors either need to provide additional experimental data to support the activity of specific regions as centromeres and need to introduce the relevance of Copia elements, as demonstrated previously, to mark centromeric regions in Phytophthora.

Third, I have several comments related to grammar/spelling that need to be addressed:

L52: '...a lot of...' too colloquial, please rephrase

L66: '...which are also evolving rapidly regions' grammatically not correct, please correct

L67: '... high quality fungal genomes...' genomes are not high quality, but genome assemblies are. please correct

L235: 'CF is of great significance.' significance for what? join with next sentence and/or rephrase for clarity

L249: 'Fluctuations'  'Variation in'

L494: Additional information is required. Which software was used, which setting? (see also comments above)

L604: I think the a clear verb is missing in this sentence 'are present' 'occur in' or similar, please rephrase

L702: 'fluctuating'  'varying'

L711: '...had more chromosome numbers...'  '...had higher chromosome numbers...'

L720 '...basis..'  '...mechanisms...'

L722: '...', contributing to understanding of karyotype evolution' this should be deleted; grammatically not correct w.r.t. to the remainder of the sentence

L766: 'Neither the intact Copia-like nor Gypsy transposons have no significant difference between fused and non-fused chromosomes (Supplementary Fig. 9),...' the grammar in this sentence seems to be incorrect (double negation) and 'have no significant difference' should be 'show' or 'display'

L956: 'ANK members were classified into five groups, in which the secreted ANKs of oomycetes was uniquely expanded in a group' what does 'in a group' refer to here? please remove or clarify

L979: 'The orderly distribution of ANK, RxLR, CRN, LPMO, and GH12 family effectors in tandem arrays in adjacent regions of the genome indicated that there may be an evolutionary and/or functional correlation.' what is an 'orderly distribution' could this please be explained?

L1189: 'A similar mechanism occurs between the core chromosome and the accessory chromosome.' it is not clear from the manuscript what this sentence refers to. Is this in *P. sojae*, or does this refer to other plant pathogens where similar mechanisms have been observed? Please clarify and add appropriate references

L1225: '...family in oomycetes, promoting the research progress of secreted ANKs.'

L1230: the gene name should be italic

L1638: 'The internal lines of chromosomes represent collinearity.' not clear what the internal lines refer to. The ribbons connecting the chromosomes? please clarify. On the same figure, the figure caption for panel b is confusing and needs clarification.

L1708: 'The number in circle represents'  'The number in THE circle represents'

L1710: 'The location of ANK repeats, LPMOs, RxLR, CRN, GH12 (Glycosyl hydrolase family 12), and RT (Retrotransposon) on chromosomes.' The sentence misses a verb, please correct

We appreciate the positive feedback from the editor and reviewers. Our detailed responses to the further suggestions for revision are as follows.

REVIEWER COMMENTS

Reviewer #1 (Remarks to the Author):

Dear authors,

Thanks for your in-depth revisions. I am happy I could help improving the manuscript.

Best regards,

Reviewer #1 aka Marco Thines

Thank you very much for your constructive feedback and guidance throughout the revision process. We deeply appreciate the time and effort you dedicated to improving our manuscript. We are glad to hear that you are satisfied with the final outcome and feel grateful for your help.

Reviewer #2 (Remarks to the Author):

The revised manuscript by Zhang and colleagues successfully addresses most of my concerns raised by the initial submission. I believe that the submission has greatly improved in terms of robustness and clarity, especially acknowledging previous work. The submitted manuscript reports on comprehensive and high-quality comparative genomic analyses to study chromosomal structures in several oomycetes and to link large-scale chromosomal variation to the evolution of virulence - I anticipate that the study will be of high relevance for scientist working on oomycetes and other plant pathogens, as well as scientist interested in genome evolution in general.

Thank you for your positive feedback on our revised manuscript. We hope our study can provide meaningful insights for scientists in related fields.

I however have a few remaining concerns/comments that have arisen from the revisions that necessitates the authors' attention.

Q1: First, based on a very valid comments raised by reviewer 1 (Q10), the manuscript now reports a novel phylogenetic analyses (Fig 2a). It is important to note that the relationship of downy mildews within Peronosporales is a sensitive topic in the research community with different and often conflicting data being reported over the years. While this is not the main topic of this article, the relationships between species are relevant for reconstructing ancestral genomes and trace evolution. While the here reported monophyly of downy mildews, based on few marker genes, seems plausible, the authors have data at hand to further corroborate these by analyzing a larger set of genes (e.g., all single-copy orthologues) with alternative methods such as ML and ME.

R1: Thanks for your suggestions. Using all single-copy genes to construct a tree is indeed more plausible. Here, we reconstructed the phylogenetic tree, incorporating all 24 single-copy genes across 38 species. Following the previous workflow of alignment, trimming, and tree construction, the reconstructed tree topology displays the monophyly of downy mildews. This updated result is now illustrated in **Fig. 2a**.

Q2: Moreover, the authors need to directly assess the impact of alternative tree topologies to the ancestral chromosome reconstruction and the number of evolutionary events necessary to reconcile the karyotypes and their evolution, given a specific phylogeny. This would not only provide a robust assessment of the karyotype evolution but might also provide additional data towards the ongoing discussion on the relationships of downy mildews.

R2: Thank you for your suggestions. Here, we first generated two different tree topologies. Subsequently, for both phylogenetic trees, we reconstructed the ancestral chromosomes and assessed evolutionary events separately. We hope these results can provide additional data and insights into the ongoing discussion about the phylogenetic relationships of downy mildews. As more chromosome-level genome data become available, we anticipate being able to perform even more precise reconstructions of ancestral chromosomes in the future. Based on the tree reconstructed from the 24 single-copy gene set, we have updated **Fig. 2** and presented additional

results in **Supplementary Fig. 7**. We have added content on the assessment of karyotype evolution in the discussion section (**L465-473**). The alternative tree topology is shown below.

Q3: Second, the manuscript reports on the centromeric regions. The manuscript notes that ‘..., and the excessive centromeres on the fused chromosomes become inactivated. This was manifested by the absence of Copia-like sequences in these regions (Supplementary Fig. 8).’. The inactivation is not experimentally shown, and the potential relevance of Copia elements to mark potential centromeres has not been introduced. To support this firm statement, the authors either need to provide additional experimental data to support the activity of specific regions as centromeres and need to introduce the relevance of Copia elements, as demonstrated previously, to mark centromeric regions in *Phytophthora*.

R3: This evidence is indeed essential. We have further confirmed the relevance of *Copia* elements to the centromeric regions in *Phytophthora sojae* using ChIP-seq data from Fang et al. This data demonstrates that *Copia*-like transposons are highly enriched in the regions marked by the centromeric histone CENP-A. The statement regarding ‘inactivated’ seems to be ambiguous; we have revised it to ‘the excessive centromeres on the fused chromosomes may be lost’. The results are presented in **Supplementary Fig. 1**.

The reference list is below:

- (1) Fang Y, Coelho MA, Shu H, Schotanus K, Thimmappa BC, et al. (2020) Long transposon-rich centromeres in an oomycete reveal divergence of centromere features in Stramenopila-Alveolata-Rhizaria lineages. *PLoS Genetics* 16(3): e1008646.

Third, I have several comments related to grammar/spelling that need to be addressed:

Q4: L52: ‘...a lot of...’ too colloquial, please rephrase

R4: Corrected. The sentence has been rephrased to ‘... causing significant economic losses.’

Q5: L66: ‘...which are also evolving rapidly regions’ grammatically not correct, please correct

R5: Corrected. The sentence has been rephrased to ‘... which are also rapidly evolving regions.’

Q6: L67: ‘... high quality fungal genomes...’ genomes are not high quality, but genome assemblies are. please correct

R6: Corrected.

Q7: L235: ‘CF is of great significance.’ significance for what? join with next sentence and/or rephrase for clarity

R7: Corrected. The sentence has been omitted to enhance clarity.

Q8: L249: ‘Fluctuations’  ‘Variation in’

R8: Corrected.

Q9: L494: Additional information is required. Which software was used, which setting? (see also comments above)

R9: Corrected. Detailed parameter settings are displayed in Supplementary Methods 1 and 2.

Q10: L604: I think a clear verb is missing in this sentence ‘are present’ ‘occur in’ or similar, please rephrase

R10: Corrected.

Q11: L702: ‘fluctuating’  ‘varying’

R11: Corrected.

Q12: L711: ‘...had more chromosome numbers...’  ‘...had higher chromosome numbers...’

R12: Corrected.

Q13: L720 ‘...basis...’  ‘...mechanisms...’

R13: Corrected.

Q14: L722: ‘..., contributing to understanding of karyotype evolution’ this should be deleted; grammatically not correct w.r.t. to the remainder of the sentence

R14: Corrected.

Q15: L766: ‘Neither the intact *Copia*-like nor *Gypsy* transposons have no significant difference between fused and non-fused chromosomes (Supplementary Fig. 9), ...’ the grammar in this sentence seems to be incorrect (double negation) and ‘have no significant difference’ should be ‘show’ or ‘display’

R15: Corrected. The sentence has been rephrased to ‘... show significant differences between ...’

Q16: L956: ‘ANK members were classified into five groups, in which the secreted ANKs of oomycetes was uniquely expanded in a group’ what does ‘in a group’ refer to here? please remove or clarify

R16: Corrected.

Q17: L979: ‘The orderly distribution of ANK, RxLR, CRN, LPMO, and GH12 family effectors in tandem arrays in adjacent regions of the genome indicated that there may be an evolutionary and/or functional correlation.’ what is an ‘orderly distribution’ could this please be explained?

R17: Corrected. The sentence was inappropriate; for clarity, it has been rephrased to ‘The tandem arrangement of ANK, RxLR, CRN, LPMO, and GH12 family effectors in genomic regions indicated ...’

Q18: L1189: ‘A similar mechanism occurs between the core chromosome and the accessory chromosome.’ it is not clear from the manuscript what this sentence refers to. Is this in *P. sojae*, or does this refer to other plant pathogens where similar mechanisms have been observed? Please clarify and add appropriate references

R18: Corrected. ‘... similar mechanism occurs between the core chromosome and the accessory chromosome in wheat blast fungus¹⁴ (L459)

Q19: L1225: ‘...family in oomycetes, promoting the research progress of secreted ANKs.’

R19: Corrected.

Q20: L1230: the gene name should be italic

R20: Corrected. In this manuscript, gene IDs are indicated in italics, while protein IDs are presented in standard font.

Q21: L1638: ‘The internal lines of chromosomes represent collinearity.’ not clear what the internal lines refer to. The ribbons connecting the chromosomes? please clarify. On the same figure, the figure caption for panel b is confusing and needs clarification.

R21: Corrected. The sentence was confusing and has been rephrased to ‘The grey lines connecting chromosomes represent regions of conserved synteny.’ The details of the modifications were displayed in the revised manuscript (L843-847).

Q22: L1708: ‘The number in circle represents’  ‘The number in THE circle represents’

R22: Corrected.

Q23: L1710: ‘The location of ANK repeats, LPMOs, RxLR, CRN, GH12 (Glycosyl hydrolase family 12), and RT (Retrotransposon) on chromosomes.’ The sentence misses a verb, please correct.

R23: The sentence has been rephrased to ‘..., and RT (Retrotransposon) is shown on chromosomes.’

Once again, we thank the editor and the two reviewers for their suggestions. We are deeply grateful to Reviewer #1 and Reviewer #2 for the time and effort they have spent on improving our manuscript. We also appreciate Reviewer #2 for correcting our errors and providing valuable new insights, which have undoubtedly enhanced the quality of our work further.

REVIEWERS' COMMENTS

Reviewer #2 (Remarks to the Author):

The revised manuscript by Zhang and colleagues successfully addresses most of my concerns raised by the initial submission. I believe that the submission has greatly improved. I also appreciate the authors' efforts to address my remaining comment on the chromosome-reconstruction based on alternative topologies.

My only remaining suggestion is directly linked to the alternative reconstructions. Based on the observed chromosome structure and the number of fusions/fissions, could the authors please speculate which topology leads to more parsimonious reconstruction, and by extension might be better supported. Adding a sentence or two to the discussion would suffice to address this point.

REVIEWERS' COMMENTS

Reviewer #2 (Remarks to the Author):

The revised manuscript by Zhang and colleagues successfully addresses most of my concerns raised by the initial submission. I believe that the submission has greatly improved. I also appreciate the authors' efforts to address my remaining comment on the chromosome-reconstruction based on alternative topologies.

Thanks for your recognition.

Q1: My only remaining suggestion is directly linked to the alternative reconstructions. Based on the observed chromosome structure and the number of fusions/fissions, could the authors please speculate which topology leads to more parsimonious reconstruction, and by extension might be better supported. Adding a sentence or two to the discussion would suffice to address this point.

R1: Thanks for your suggestions. In the revised 'Discussion' section, we added a statement to support our hypothesis: 'Karyotype reconstructions based on different tree topologies show slight variations, linked not only to differences in the topology of phylogenetic trees but also to the impact of limited chromosome-level assemblies. Further chromosome-level assemblies and exploration of the relationships of downy mildews within Peronosporales remain necessary. Reconstruction using the topology tree shown in **Fig. 2a** appears to better fit the results of chromosome synteny. However, the fragmented nature of assemblies for most species may disrupt continuous ancestral regions without enough supporting evidence.'